# Fitness effects of CRISPR endonucleases in *Drosophila melanogaster* populations

Anna M Langmüller[1,2,3†], Jackson Champer[1,4,5]*[†], Sandra Lapinska[1,4], Lin Xie[1,4], Matthew Metzloff[1,4], Samuel E Champer[1], Jingxian Liu[1,4], Yineng Xu[1,4], Jie Du[5], Andrew G Clark[1,4], Philipp W Messer[1]*

[1]Department of Computational Biology, Cornell University, Ithaca, United States; [2]Institut für Populationsgenetik, Vetmeduni Vienna, Vienna, Austria; [3]Vienna Graduate School of Population Genetics, Vetmeduni Vienna, Vienna, Austria; [4]Department of Molecular Biology and Genetics, Cornell University, Ithaca, United States; [5]Center for Bioinformatics, School of Life Sciences, Peking-Tsinghua Center for Life Sciences, Peking University, Beijing, China

**Abstract** Clustered Regularly Interspaced Short Palindromic Repeats (CRISPR)/Cas9 provides a highly efficient and flexible genome editing technology with numerous potential applications ranging from gene therapy to population control. Some proposed applications involve the integration of CRISPR/Cas9 endonucleases into an organism's genome, which raises questions about potentially harmful effects to the transgenic individuals. One example for which this is particularly relevant are CRISPR-based gene drives conceived for the genetic alteration of entire populations. The performance of such drives can strongly depend on fitness costs experienced by drive carriers, yet relatively little is known about the magnitude and causes of these costs. Here, we assess the fitness effects of genomic CRISPR/Cas9 expression in *Drosophila melanogaster* cage populations by tracking allele frequencies of four different transgenic constructs that allow us to disentangle 'direct' fitness costs due to the integration, expression, and target-site activity of Cas9, from fitness costs due to potential off-target cleavage. Using a maximum likelihood framework, we find that a model with no direct fitness costs but moderate costs due to off-target effects fits our cage data best. Consistent with this, we do not observe fitness costs for a construct with Cas9HF1, a high-fidelity version of Cas9. We further demonstrate that using Cas9HF1 instead of standard Cas9 in a homing drive achieves similar drive conversion efficiency. These results suggest that gene drives should be designed with high-fidelity endonucleases and may have implications for other applications that involve genomic integration of CRISPR endonucleases.

*For correspondence:
jchamper@pku.edu.cn (JC);
messer@cornell.edu (PWM)

†These authors contributed equally to this work

## Editor's evaluation

The manuscript describes an attempt to assess fitness costs of CRISPR/Cas9 endonucleases in the context of gene drive in *Drosophila melanogaster* by looking at direct fitness costs of the transgene and indirect fitness costs due to off-target cleavage. The authors performed experimental cage population studies and a maximum-likelihood approach to disentangle the contribution of direct and off-target-related fitness costs. The combined experimental and mathematical approach allows the authors to conclude that off-target cleavage is largely responsible for the observed fitness costs, although no mutated alleles were detected at the most likely computational predicted off-target sites. The authors also use a high-fidelity Cas9 nuclease (Cas9HF) to confirm reduced fitness costs probably due to increased cleavage specificity. The data are of interest for CRISPR/Cas9 applications in general and for gene drive applications in particular and the manuscript is of interest to a wide range of readers.

## Introduction

The ability to make specific edits of genetic material has been a long-standing goal in molecular biology. Until recently, such DNA engineering was cumbersome, expensive, and difficult since it relied on site-specific nucleases or random insertions. CRISPR technology represents a milestone in genome editing because it makes DNA engineering highly efficient, relatively simple to use, and cost-effective through the use of endonucleases that can be flexibly programmed to cut specific sequences dictated by a guide RNA (gRNA; *Moon et al., 2019*; *Mali et al., 2013*).

The programmability of CRISPR/Cas9 systems allows for numerous potential applications (*Moon et al., 2019*), including cancer and disease treatment (*Chen et al., 2019*; *Hodges and Conlon, 2019*; *Jo et al., 2019*; *Zhang et al., 2020a*; *Yang et al., 2017*), stimuli tracking in living cells (*Tang and Liu, 2018*), and crop improvement (*Zhang et al., 2020b*). While most applications of CRISPR use this technology to engineer specific modifications in a given DNA sequence, some proposed applications take the idea one step further by integrating the CRISPR machinery itself into an organism's genome. In that case, endonuclease activity can continue to produce genetic changes in the cells of the living organism. When present in the germline, these genetic changes might even be passed on to future generations, such as in CRISPR-based gene drives—'selfish' genetic elements that are engineered to rapidly spread a desired genetic trait through a population (*Esvelt et al., 2014*; *Champer et al., 2016*; *Unckless et al., 2017*; *Noble et al., 2017*; *Burt, 2014*).

However, major questions loom large about the technical feasibility of these proposed applications. For example, it remains unclear whether activity of CRISPR endonucleases could entail unintended and potentially harmful consequences in the transgenic organisms, for instance due to the tendency to produce non-specific DNA modifications (so-called 'off-target effects'; *Zhang et al., 2015*). Such off-target cleavage could be substantially higher when Cas9 is continuously expressed from a genome and inherited by offspring, where further off-target cleavage can occur.

In this study, we seek to address this question in the context of CRISPR gene drive, an emerging technology that could be used for applications ranging from the control of vector-borne diseases to the suppression of invasive species (*Esvelt et al., 2014*; *Unckless et al., 2017*; *Burt, 2014*; *Alphey, 2014*). One class of CRISPR-based gene drives is the so-called 'homing drives'. These genetic constructs are programmed to cleave a wild-type sister chromatid and get copied to the target site through homology-directed repair. Since 'homing' occurs in the germline, the drive allele will be inherited at a super-Mendelian rate and can thereby spread quickly through the population. The effectiveness of such systems has now been demonstrated in various organisms, including yeast (*Roggenkamp et al., 2018*; *Shapiro et al., 2018*; *DiCarlo et al., 2015*; *Basgall et al., 2018*), mosquitoes (*Hammond et al., 2017*; *Gantz et al., 2015*; *Kyrou et al., 2018*; *Hammond et al., 2016*), fruit flies (*Oberhofer et al., 2018*; *Carrami et al., 2018*; *Gantz and Bier, 2015*; *Champer et al., 2017*; *Champer et al., 2018*; *Champer et al., 2019a*; *Champer et al., 2019b*; *Champer et al., 2020b*), and mice (*Grunwald et al., 2019*). Another class of CRISPR gene drives operates by the 'toxin-antidote' principle (*Champer et al., 2020a*). Here, the drive allele serves as the 'toxin' by carrying a CRISPR endonuclease programmed to target and disrupt an essential wild-type gene. At the same time, the construct also contains a recoded version of that gene (the 'antidote'), which is immune to cleavage by the drive. Over time, the drive will continuously increase in frequency and remove wild-type alleles from the population (*Burt and Crisanti, 2018*). Both homing and toxin-antidote drives can be 'modification drives', intended to spread a desired genetic payload through the population (e.g. a gene that prevents mosquitoes from transmitting malaria), or 'suppression drives' that seek to diminish or outright eliminate the target population (*Champer et al., 2020a*; *Champer et al., 2021a*).

A key factor in determining the expected population dynamics of any type of gene drive is the fitness cost imposed by the drive (*Wedell et al., 2019*). Such fitness costs could come in the form of reduced viability, fecundity, or mating success of the individuals that carry drive alleles. In suppression drives, some fitness costs are typically an intended feature of the drive, necessary to ultimately achieve population suppression. However, these costs are usually recessive to allow the drive to spread to high frequency, and there is generally a limit as to how high other costs can be before the drive will lose its ability to spread effectively (*Champer et al., 2020a*; *Champer et al., 2021a*; *Champer et al., 2021b*; *Deredec et al., 2008*). For modification drives, fitness costs tend to slow the spread of the drive and can thereby increase the chance that resistance alleles evolve, which could ultimately defeat the drive (*Unckless et al., 2017*). For such applications, it is therefore desirable to minimize any fitness costs.

In drives with frequency-dependent invasion dynamics, such as most CRISPR toxin-antidote systems (*Champer et al., 2020b*; *Oberhofer et al., 2019*), fitness costs typically determine the frequency threshold required for the drive to spread through the population (*Champer et al., 2020a*; *Champer et al., 2021a*; *Champer et al., 2021b*).

We believe it is useful to distinguish between two types of fitness costs of a gene drive. The first class comprises any costs resulting from the genomic integration of the drive construct (e.g. when this disrupts a functionally important region), costs of potential 'payload' genes included in the drive construct, costs resulting directly from the expression of the endonuclease or other drive elements such as gRNAs, and costs due to cleavage of the intended target site. We will call these 'direct' fitness costs. By contrast, the second class comprises any potential fitness costs resulting from cleavage and disruption of unintended sites in the genome, so-called 'off-target' effects. Despite their critical importance, we still know surprisingly little about the specific types of fitness costs imposed by gene drives. Furthermore, it remains unclear whether there are certain baseline fitness costs that would be difficult to avoid in any gene drive construct, for instance, because they are inherent to the expression and activity of the CRISPR endonuclease.

In this study, we conduct a comprehensive assessment of the fitness effects resulting from the genomic expression of CRISPR/Cas9 in experimental *Drosophila melanogaster* populations. We specifically investigate four different transgenic constructs that allow us to disentangle direct fitness costs from those due to off-target effects. We estimate these fitness costs both through statistical inference

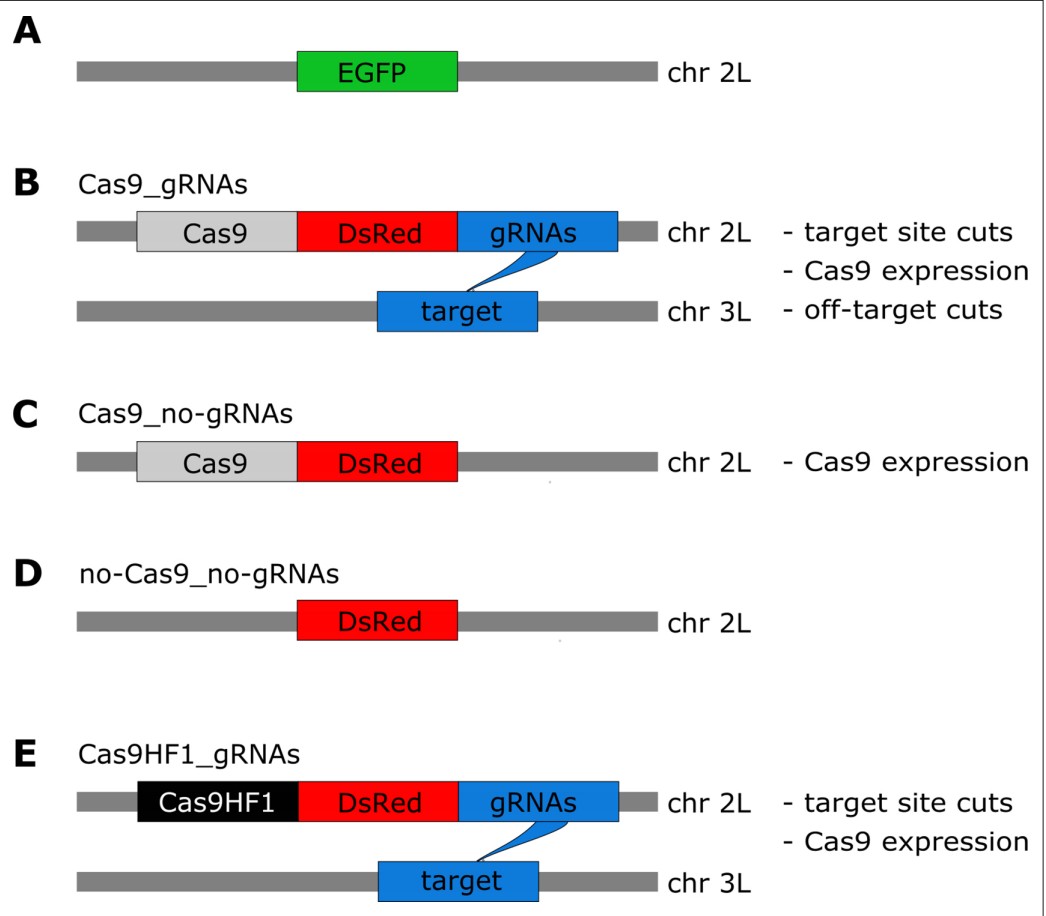

**Figure 1.** Overview of constructs and the potential types of fitness costs in the four constructs. (**A**) The starting point for our constructs is an EGFP marker inserted into chromosome 2 L (~20.4 Mb). The four constructs are then inserted into this EGFP locus (thereby disrupting EGFP). (**B**) The Cas9_gRNAs construct contains Cas9, DsRed, and gRNAs. The gRNAs target chromosome 3 L (~18.3 Mb), instead of the sister chromatid. (**C**) The Cas9_no-gRNAs construct carries Cas9 and DsRed, but no gRNAs are expressed. (**D**) The no-Cas9_no-gRNAs construct carries only the fluorescent marker DsRed. (**E**) The Cas9HF1_gRNAs construct has the same structure as Cas9_gRNAs but carries Cas9HF1 instead of Cas9.

from allele frequency trajectories in cage populations using a maximum likelihood approach, and a direct evaluation of individual fitness components using viability, fecundity, and mate choice assays.

## Results

### Construct design

We designed four constructs to assess the fitness costs of *in vivo* CRISPR/Cas9 expression in *D. melanogaster*. As a starting point for our transgenic fly lines, we engineered an Enhanced Green Fluorescent Protein (EGFP) marker driven by the 3xP3 promoter into a gene-free, nonheterochromatic position on chromosome 2 L (region targeted by gRNA: 20,368,542–20,368,561; *Figure 1A*). This EGFP marker was then used as an insertion point for the four constructs we tested. Our first construct, 'Cas9_gRNAs', contains Cas9 expressed by the *nanos* promoter, the fluorescence marker *Discosoma* sp Red (DsRed) driven by the 3xP3 promoter, and four gRNAs driven by the U6:3 promoter (*Figure 1B*), which are separated by tRNAs that are removed after transcription (*Champer et al., 2018*). The gRNAs of the Cas9_gRNAs construct target a gene-free, nonheterochromatic position on a different chromosome (3 L, region targeted by gRNAs: 18,297,270–18,297,466), preventing any homing activity. In addition to Cas9_gRNAs, three other constructs were designed: 'Cas9_no-gRNAs' has a similar architecture as Cas9_gRNAs but lacks the four gRNAs driven by the U6:3 promoter (*Figure 1C*) 'no-Cas9_no-gRNAs' contains neither Cas9 nor the gRNAs but only the fluorescence marker DsRed driven by the 3xP3 promoter (*Figure 1D*) the last construct, 'Cas9HF1_gRNAs' (*Figure 1E*), has the same architecture as Cas9_gRNAs, except that Cas9 is replaced by a high-fidelity version (Cas9HF1), which has been reported to largely eliminate off-target cleavage (*Kleinstiver et al., 2016*). We confirmed with PCR-based genotyping that—as expected—all progeny of individuals with the Cas9_gRNAs and Cas9HF1_gRNAs alleles had at least one of their gRNA target sites mutated and that all four gRNAs were similarly active in both these constructs.

The specific designs of these four constructs allow us to identify and disentangle different types of Cas9-related fitness costs. If double-strand breaks at the target site impose fitness costs, such costs should be present for the Cas9_gRNAs and Cas9HF1_gRNAs constructs, but not for the Cas9_no-gRNAs and no-Cas9_no-gRNAs constructs. Cas9_no-gRNAs have no gRNAs expressed to guide Cas9 to the target site and without gRNAs, Cas9 does not cleave DNA (*Jinek et al., 2012*; *Cong et al., 2013*). The no-Cas9_no-gRNAs construct neither expresses Cas9 nor the gRNAs. If the expression of Cas9 imposes a fitness cost, all constructs except for no-Cas9_no-gRNAs should incur such a cost, because only this construct does not express Cas9. If off-target effects of Cas9 impose fitness costs, only the Cas9_gRNAs construct should incur them, because the designs of Cas9_no-gRNAs and no-Cas9_no-gRNAs prevent cutting events, and Cas9HF1_gRNAs reportedly have a much lower rate of off-target cleavage (*Kleinstiver et al., 2016*). *Figure 1* summarizes the designs and different potential fitness costs for our four constructs.

### Population cage experiments

To assess the fitness effects of the four constructs, we tracked their population frequencies relative to the baseline EGFP construct (*Figure 1A*) over several generations in large cage populations by phenotyping the whole population for both dominant fluorescent markers (DsRed and EGFP). Overall, we assessed 13 cages: seven with the Cas9_gRNAs construct, and two each with the Cas9_no-gRNAs, no-Cas9_no-gRNAs, and Cas9HF1_gRNAs constructs (*Figure 2*). In each cage population, the construct frequency was tracked for at least eight consecutive, non-overlapping generations. The median population size across all experiments was 3602 (*Figure 2—figure supplement 1*). To avoid potentially confounding maternal fitness effects on the construct frequency dynamics (which could arise based on minor differences in health or age between the initial batches of flies mixed together), we excluded the first generation of five cage populations (Cas9_gRNAs construct: replicates 1, 2, 5, 6, and 7) from the analysis, because their founding individuals (construct homozygotes and EGFP homozygotes) were raised in potentially different environments.

The Cas9_gRNAs construct was the only one that systematically decreased in frequency over the course of the experiment (average allele frequency change = –0.11, SEM = 0.03, *Figure 2*). This provides a first indication that Cas9 off-target effects could be the primary driver of fitness costs. However, the frequency dynamics of the Cas9_gRNAs construct varied widely between individual

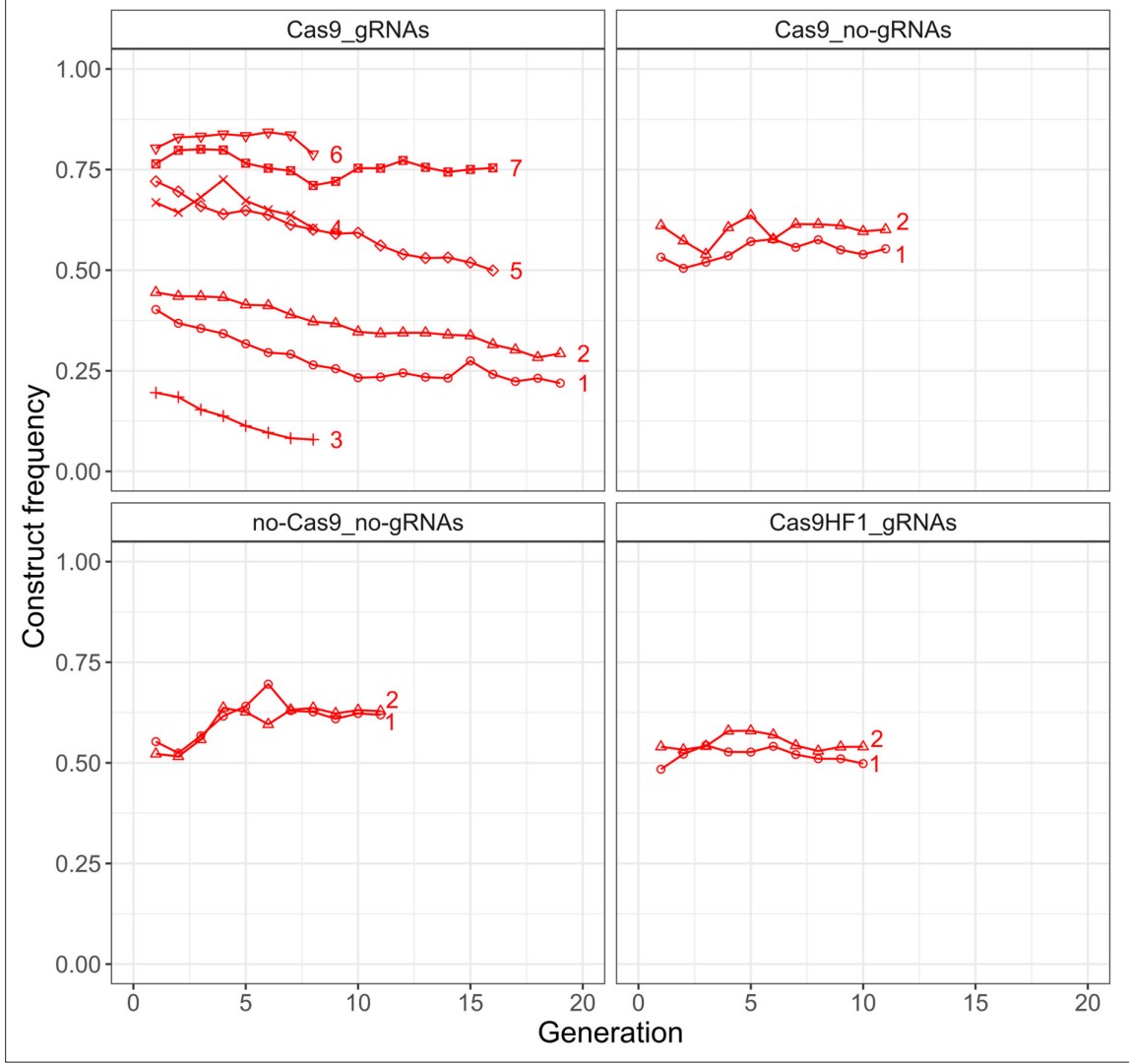

**Figure 2.** Construct frequency trajectories in the cage populations. Each line is one cage experiment. To obtain construct frequencies, we screened all adult flies for each generation in the respective cage experiments (see *Figure 2—figure supplement 1* for population sizes).

The online version of this article includes the following figure supplement(s) for figure 2:

**Figure supplement 1.** Population sizes of all *D. melanogaster* cage populations.

cage populations (*Figure 2*). For example, in the two replicates where the construct had the highest starting frequency, its frequency remained approximately constant, whereas it clearly decreased in the other replicates.

## Maximum likelihood analysis

To provide a more quantitative analysis of the fitness costs of the different constructs in our cage populations, we employed a maximum likelihood framework developed for the estimation of selection parameters based on genotype frequency time series data (*Liu et al., 2019*). We specifically modified the method to support two unlinked autosomal loci, representing the construct and a single idealized off-target site (see the section on 'Maximum likelihood framework for fitness cost estimation' in the Methods for a more detailed description of the underlying model). This model can estimate fitness costs with CI while fully accounting for stochastic allele frequency fluctuations due to random genetic drift. Furthermore, we can perform statistical model selection and goodness-of-fit analyses on different selection scenarios to disentangle different types of fitness costs for each construct.

## General model assumptions

Each of the two loci in our model is biallelic (EGFP/construct; uncut/cut off-target site). In individuals that carry a construct, all uncut off-target alleles are assumed to be cut in the germline (i.e. germline cut rate was set to 1), which are then passed on to offspring that could suffer negative fitness consequences. In the early embryo, all uncut off-target alleles are assumed to be cut by maternally deposited Cas9/gRNAs if the mother carries at least one construct allele (i.e. embryo cut rate was also set to 1), changing the individual's genotype at the off-target site and exposing it to the potential fitness costs associated with this new genotype. Because individuals could carry numerous off-target sites, and the fitness of cleaved alleles could differ vastly between off-target sites, our model of a single off-target site is highly idealized. However, modeling a more complex off-target landscape would require numerous parameters (fitness costs, cut rates, epistatic interactions, etc.) that would be difficult if not impossible to disentangle given our limited number of data points. To reduce model complexity, we therefore limited the model to one off-target locus being always cut in the presence of Cas9.

Fitness costs due to carrying the construct and/or the presence of cut off-target sites are assumed to be multiplicative across the two loci, as well as for the two alleles at each locus. We studied models where fitness costs affect only viability, and models where they affect only mate choice and fecundity (both equally). Overall, our maximum likelihood model infers three parameters: the effective population size ($N_e$) of the cage, the 'direct fitness estimate' of the construct (defined as the relative fitness of construct/EGFP heterozygotes versus EGFP/EGFP homozygotes), and the 'off-target fitness estimate' (defined as the relative fitness of cut/uncut heterozygotes versus uncut/uncut homozygotes). Note that in our idealized model with a single cleavage site, this site could in principle also represent 'on-target' cleavage. However, due to the intergenic location of all gRNA target sites in our constructs, we do not expect such fitness costs to be present. Furthermore, if on-target cleavage had a measurable negative fitness effect, this should have been apparent in the frequency trajectories of the Cas9HF1_gRNAs construct. Since this construct had no apparent reduction in fitness (**Figure 2**, Table 2), we refer to this fitness parameter exclusively as 'off-target'.

## Model evaluation

For each construct, five different models with different selection scenarios were studied (**Table 1**): in the 'full inference model', both the construct and cut off-target alleles can impose fitness costs. In the 'construct' model, only construct alleles impose a fitness cost. In the 'off-target' model, only cut off-target alleles impose a fitness cost. In the 'initial off-target model', we assumed that fitness costs originated before the experiment (e.g. through the injection process or perhaps transient maternal effects in the ancestral generation). For the 'initial off-target model', the construct homozygotes in the ancestral generation all had cut off-target alleles, but no additional off-target cutting occurred during the experiment (i.e. the germline and embryo cut rate were set to 0). Finally, in the 'neutral' model, no fitness costs were present at all.

Inferences were performed on the combined data of the replicated experimental populations for each construct. The individual models were compared using the corrected Akaike information criterion (AICc; **Akaike, 1998**)—a goodness-of-fit measure that also penalizes for complexity (i.e. number of parameters) in a given model. A lower AICc value indicates a higher quality model.

**Table 1.** Fitness cost model overview.
The table shows which types of fitness costs are contained in each model.

| Model | Construct allele | Cut off-target allele |
|---|---|---|
| Full inference | + | + |
| Construct | + | − |
| Off-target | − | + |
| Initial off-target | − | +* |
| Neutral | − | − |

*No additional cutting events at off-target sites during the experiment.

## Construct frequency dynamics match a model with moderate off-target fitness costs

For the Cas9_gRNAs construct, we found that the full inference model with viability selection yielded the highest quality, with a 'direct fitness estimate' of 0.98 and an 'off-target fitness estimate' of 0.84 (**Table 2**). Note, however, that the 95% CI of the direct fitness estimate includes a value of 1. The simpler 'off-target' model (where only cut off-target alleles impose a fitness cost) with viability

**Table 2.** Model comparison and parameter estimates for Cas9_gRNAs.

| Model | Selection | $\hat{N}_e$ | Direct fitness estimate | Off-target fitness estimate | $\ln\hat{L}$ | P | AICc |
|---|---|---|---|---|---|---|---|
| Full | Viability | 175 [140–215] | 0.98 [0.95–1.00] | 0.84 [0.77–0.91] | 384.7 | 3 | −763 |
| Full | Mate choice = fecundity | 163 [131–200] | 0.96 [0.94–0.98] | 1.00 [0.95–1.06] | 378.8 | 3 | −751 |
| Construct | Viability | 164 [131–201] | 0.96 [0.93–0.98] | 1* | 378.9 | 2 | −754 |
| Construct | Mate choice = fecundity | 163 [131–200] | 0.96 [0.94–0.98] | 1* | 378.8 | 2 | −754 |
| Off-target | Viability | 173 [139–212] | 1* | 0.80 [0.74–0.88] | 383.6 | 2 | −763 |
| Off-target | Mate choice = fecundity | 157 [126–192] | 1* | 0.95 [0.90–1.01] | 375.1 | 2 | −746 |
| Initial off-target | Viability | 156 [125–191] | 1* | 0.92 [0.82–1.02] | 374.8 | 2 | −745 |
| Initial off-target | Mate choice = fecundity | 156 [125–191] | 1* | 0.96 [0.91–1.01] | 374.8 | 2 | −745 |
| Neutral | None | 154 [123–189] | 1* | 1* | 373.6 | 1 | −745 |

Each row shows the parameter estimates for effective population size ($\hat{N}_e$), maximum log likelihood ($\ln\hat{L}$) , number of free parameters in the maximum likelihood framework (*P*), and corrected Akaike information criterion value ($AICc = 2p - 2\ln\hat{L} + (2p^2 + 2p)/(n - p - 1)$) where n=87 is the number of generation transitions for a specific model and selection type. 1* entries indicate that a parameter was fixed at 1 (= no fitness effect is estimated). Values in squared brackets in the parameter estimate columns represent the 95% CI estimated from a likelihood ratio test with one degree of freedom. The model with the lowest AICc (i.e. the best fit) is highlighted in bold.

selection and direct fitness estimate set to 1 in fact had an equal AICc value to the 'full' model, which further supports that direct fitness costs in construct/EGFP heterozygotes are likely small. Models with fecundity/mate choice selection generally had lower quality than models with viability selection. The 'initial off-target' and 'neutral' models yielded the highest AICc values. Taken together, these results suggest that among the five different models we tested (*Table 1*), the observed frequency trajectories of the Cas9_gRNAs construct in our cage populations are best explained by a model where direct effects are less than a few percent and off-target effects impose moderate fitness costs of ~30% (= 1–0.84²) in cut/cut homozygotes in our idealized single off-target site model (*Table 2*).

A scenario in which fitness costs are primarily due to off-target effects also suggests a possible mechanism for why the decline in the frequency of the Cas9_gRNA construct could be greater in cages of lower construct starting frequencies, which appears to be the case in our experiment (*Figure 2*). This mechanism would work due to the accumulation of previously cut off-target sites that should typically be protected from future cutting due to sequence mutations caused by the repair process, similar to the creation of resistance alleles in a homing drive (*Champer et al., 2017*). Early in the experiment, cut off-target sites should be found primarily in individuals that also carry a construct allele. Fitness costs resulting from such cuts will therefore also impose negative selection against construct alleles. However, as mutated off-target sites accumulate over the course of an experiment, they will increasingly segregate independently from construct alleles, thereby reducing selection against these alleles. By the time all potential off-target sites in the population have been mutated, construct alleles would no longer experience any negative selection if off-target effects are indeed the only cause of fitness costs. Importantly, cages where the construct is introduced at a higher frequency (e.g. Cas9_gRNAs replicate 6 in *Figure 2*) should experience this effect faster than cages where it is introduced at a lower frequency (e.g. Cas9_gRNAs replicate 3 in *Figure 2*) due to the higher overall rate of cleavage events in the population.

To test how well our best-fitting model from *Table 1* (full inference model with viability selection) can resemble the observed frequency-dependent construct dynamics of the Cas9_gRNAs construct,

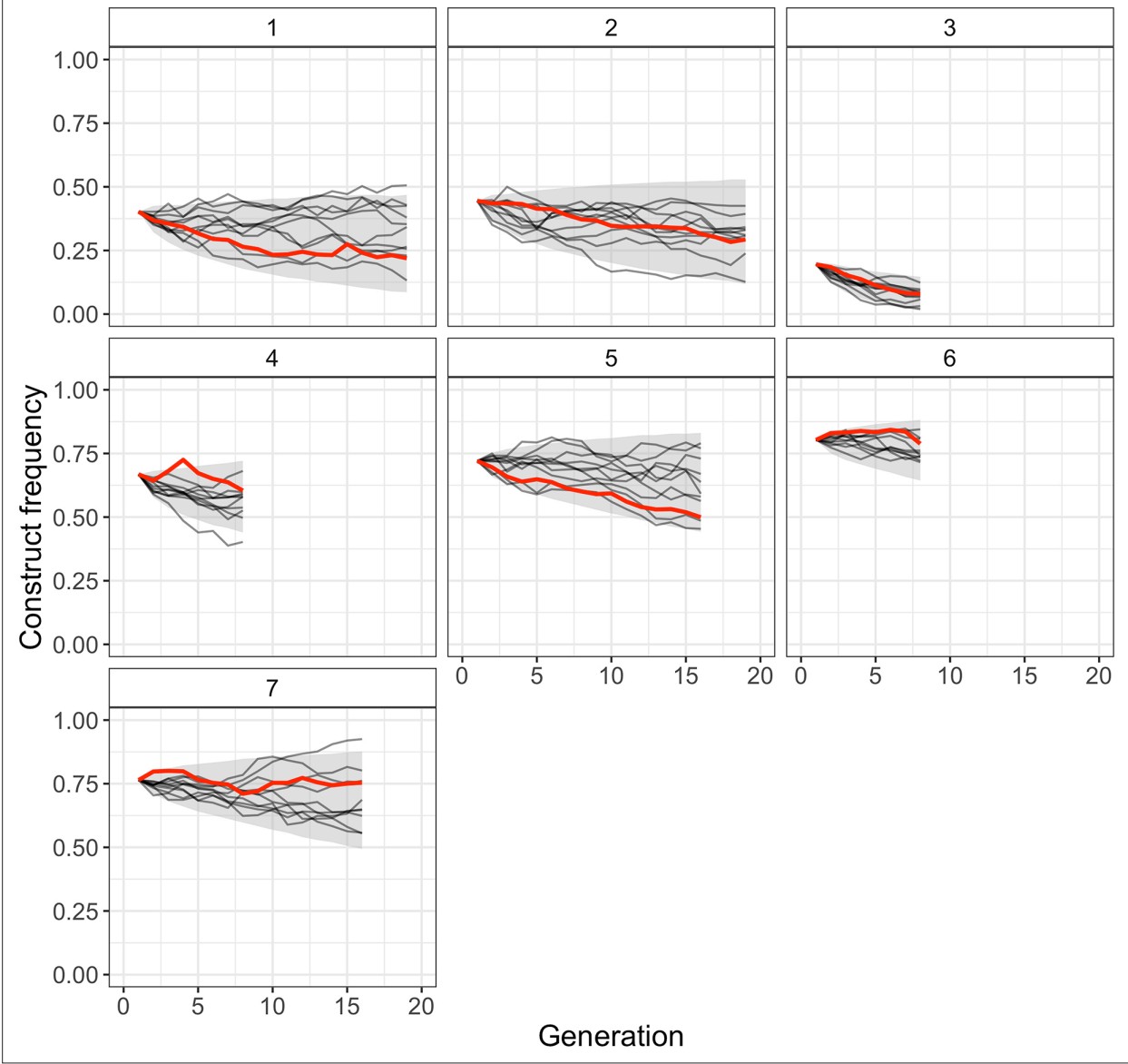

**Figure 3.** Comparison of observed Cas9_gRNAs construct frequencies with simulated trajectories of the full model with viability selection under its maximum likelihood parameter estimates ($\hat{N}_e = 175$, direct fitness estimate = 0.98, off-target fitness estimate = 0.84). Solid red lines present observed construct frequencies, black lines show 10 simulated trajectories for each cage, and the shaded area represents the range between the 2.5 and 97.5 percentile of the simulated trajectories (10,000 simulations per cage).

The online version of this article includes the following figure supplement(s) for figure 3:

**Figure supplement 1.** Comparison of observed construct frequencies (solid red line) in our experimental Cas9_gRNAs cages with the predicted trajectories of the full inference model with viability selection (dotted, orange line; off-target fitness = 0.84, direct fitness = 0.98), and the construct model with viability selection (dashed, blue line; direct fitness = 0.96), using the inferred maximum likelihood parameter estimates (**Table 2**).

**Figure supplement 2.** Comparison of observed construct frequencies with simulated trajectories of a neutral model.

we simulated construct trajectories under its maximum likelihood parameter estimates ($N_e = 175$, direct fitness estimate = 0.98, off-target fitness estimate = 0.84). We found that the simulated genotype frequencies not only closely resemble the observed decrease in construct frequency but also capture the heterogeneity in frequency trajectories observed among individual replicates (**Figure 3**). Additionally, we compared simulated trajectories for this model with simulated trajectories from the 'construct' model that only considers direct fitness costs (**Figure 3—figure supplement 1**). We found that the full inference model captures the observed frequency-dependent construct dynamics better

**Table 3.** Model comparison and parameter estimates for Cas9_no-gRNAs, no-Cas9_no-gRNAs, and Cas9HF1_gRNAs.

| Construct | Model | Selection | $\hat{N}_e$ | Direct fitness estimate | Off-target fitness estimate | $\ln\hat{L}$ | P | AICc |
|---|---|---|---|---|---|---|---|---|
| Cas9_no-gRNAs | Construct | Viability | 243 [152–366] | 1.0 [0.96–1.04] | 1* | 88.6 | 2 | –173 |
| Cas9_no-gRNAs | Initial off-target | Viability | 250 [156–377] | 1* | 0.84 [0.65–1.18] | 89.2 | 2 | –174 |
| **Cas9_no-gRNAs** | **Neutral** | **None** | **243 [152–366]** | 1* | 1* | 88.6 | 1 | –175 |
| no-Cas9_no-gRNAs | Construct | Viability | 162 [101–243] | 1.0 [0.97–1.10] | 1* | 81.5 | 2 | –158 |
| no-Cas9_no-gRNAs | Initial off-target | Viability | 162 [101–243] | 1* | 1.12 [0.84–1.63] | 81.5 | 2 | –158 |
| **no-Cas9_no-gRNAs** | **Neutral** | **None** | **162 [101–243]** | 1* | 1* | 81.5 | 1 | –161 |
| Cas9HF1_gRNAs | Construct | Viability | 396 [240–608] | 1.0 [0.97–1.04] | 1* | 88.1 | 2 | –171 |
| **Cas9HF1_gRNAs** | **Initial off-target** | **Viability** | **433 [263–655]** | 1* | 1.18 [0.99–1.45] | 89.7 | 2 | –175 |
| Cas9HF1_gRNAs | Neutral | None | 396 [240–608] | 1* | 1* | 88.1 | 1 | –174 |

Each row shows the parameter estimates for a specific construct, model, and selection type, using the same definitions as in **Table 2**. For each construct, the best-fitting model is highlighted in bold.

than this model, with most of the improvement due to better matching trajectories from cages with low starting frequencies, where off-target effects would be expected to have a more drastic impact on the relative fitness of construct-carrying individuals.

To further support our hypothesis that off-target effects may be the primary driver of fitness costs, we applied our maximum likelihood inference framework to the experimental cage data of the three other constructs (Cas9_no-gRNAs, no-Cas9_no-gRNAs, and Cas9HF1_gRNAs). Because none of these three constructs, by design, should be capable of producing substantial amounts of off-target cuts, we set the germline and embryo cut rate to 0 (i.e. no off-target alleles are cut in the presence of a construct or maternally deposited Cas9/gRNAs) and inferred viability fitness effects for the construct. Except for the 'initial off-target' model (i.e. the model in which fitness costs originated before the experiment), construct homozygotes of the ancestral population were assumed to not carry any cut off-target alleles. For Cas9_no-gRNAs, and no-Cas9_no-gRNAs, the 'neutral' model without any fitness costs explained the observed construct frequency trajectories best (**Table 3**, **Figure 3—figure supplement 2**), further corroborating the notion that Cas9 fitness costs in our experimental populations may be primarily due to off-target cuts (**Figure 1**). However, the construct frequency dynamics of Cas9HF1_gRNAs are best explained by an 'initial off-target' model, where cut off-target alleles are beneficial, closely followed by the neutral model (**Table 3**). While we cannot completely rule out that the initial construct homozygotes of Cas9HF1_gRNAs may have had some fitness advantage due to cut off-target alleles or transgenerational beneficial effects, the 95% CI for the off-target fitness parameter still includes a fitness value of 1 (i.e. no fitness effects). A putative short-term fitness advantage could also be explained by maternal effects that persisted for the first 2–3 generations. Although we do not anticipate that any other construct than Cas9_gRNAs can produce substantial off-target effects (**Figure 1**), we repeated the analysis of the three other constructs with cut rates set to 1 (i.e. off-target alleles are always cut in the presence of a construct or maternally deposited Cas9/gRNAs) and inferred viability selection, which yielded similar results (**Supplementary file 1**).

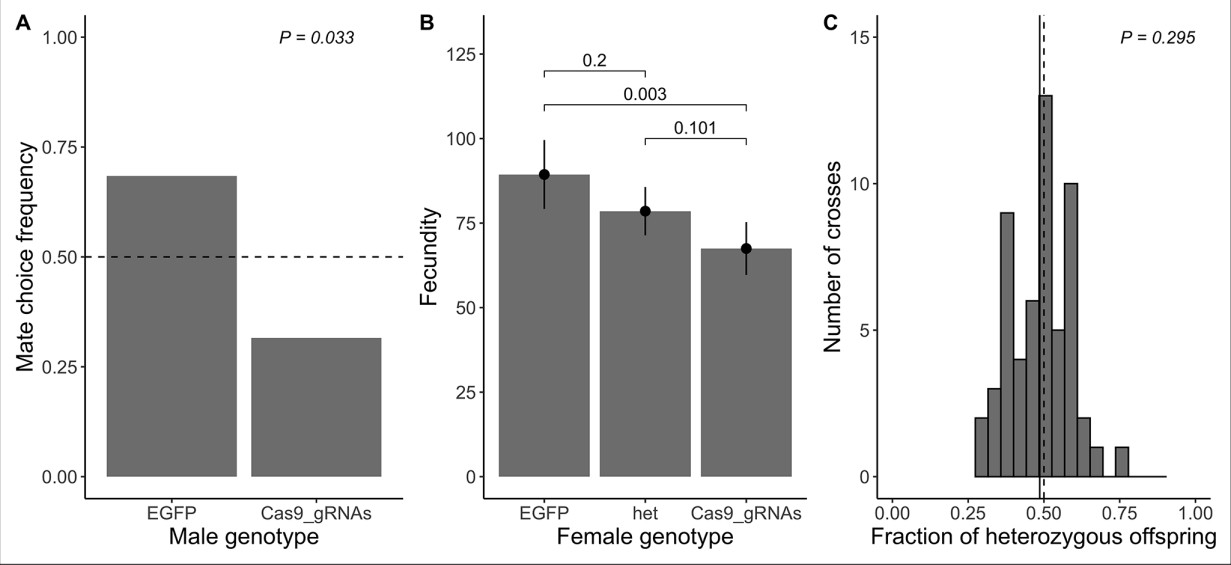

**Figure 4.** Direct measurement of fitness parameters. (**A**) Observed mate choice frequency (y-axis) of EGFP homozygous females (sample size n=38) choosing between EGFP and Cas9_gRNAs homozygous males (x-axis; as only two genotypes were tested, the frequencies sum up to 1). In case of no mate choice preference, the expected mate choice frequency is 0.5 (horizontal dashed line). The observed mate choice frequency of EGFP homozygous males was significantly different from 0.5 (exact binomial test; p=0.033; chosen level of significance α= 0.05). (**B**) Average fecundity (y-axis) for each female genotype (x-axis). The observed average fecundity, defined as the total number of eggs per female laid over the course of three consecutive days, is plotted for each female genotype separately (EGFP = EGFP homozygous females (n=27), het = heterozygous females (n=55), and Cas9_gRNAs = Cas9_gRNAs homozygous females (n=46)). All females were mated in individual crosses to EGFP homozygous males of the same age. The fitted model is shown as black dots with error bars displaying the 95% CI. The p-values of pairwise genotype comparisons adjusted with Tukey's method are displayed above the bars (α= 0.05). (**C**) Viability is measured as a fraction of heterozygous offspring from crosses between heterozygous females (n=56) and EGFP homozygous males. If the genotype does not influence viability, we expect a heterozygous offspring fraction of 0.5 (vertical dashed line). The observed fraction of heterozygous offspring (vertical solid line) does not differ from 0.5 (one sample t-test; p=0.295; α=0.05.

## Phenotypic fitness assays

As a complementary validation of the inferred fitness estimates from the maximum likelihood analysis of our cage experiments, we conducted three independent phenotypic assays (mate choice, fecundity, and viability) for flies carrying the Cas9_gRNAs construct. First, we assessed the mate choice of 40 independent EGFP homozygous females that were each set up with one EGFP homozygous and one Cas9_gRNAS homozygous male in one single vial. Of these, 38 samples had exclusively EGFP homozygous or heterozygous offspring, whereas two samples displayed offspring of both genotypes. As this suggests multiple matings of the female, we excluded these two data points from the analysis. The estimated frequency of 0.684 of EGFP homozygous females choosing EGFP homozygous males (n=26) over Cas9_gRNAs homozygous males (n=12) as mates was significantly different from 0.5 (exact binomial test p=0.033; *Figure 4A*).

Next, we measured the fecundity of 128 independent females (27 EGFP homozygotes, 55 heterozygotes, and 46 Cas9_gRNAs homozygotes), where fecundity was defined as the total number of eggs laid per female over the course of three consecutive days. Overall, the female genotype significantly influenced fecundity (full-null model comparison $F_{2,125}$=5.885, p=0.004), with Cas9_gRNAs homozygous females being significantly less fecund than EGFP homozygous females. No significant difference was detected between EGFP homozygotes and heterozygotes, or heterozygotes and Cas9_gRNAs homozygotes, respectively (*Figure 4B*).

Finally, we measured viability as the fraction of heterozygous offspring out of the total number of offsprings from 56 independent fly crosses between construct heterozygotes and EGFP homozygotes (see the section on 'Viability' in the Methods for a detailed description of the conducted fly crosses). If the genotype does not influence viability, we expect a heterozygous offspring fraction of 0.5. We observed that the fraction of heterozygous offspring was normally distributed (mean = 0.486, SD = 0.098; A=0.405, p=0.343) and did not differ significantly from 0.5 ($t_{55}$=−1.057, p=0.295; *Figure 4C*).

In summary, we found that Cas9_gRNAs homozygous males were 46.2% less likely to be picked as mates by EGFP homozygous females (*Figure 4A*), and Cas9_gRNAs homozygous females laid on average 24.5% less eggs than EGFP homozygous females (*Figure 4B*). These findings suggest that—in contrast to the maximum likelihood model assumption—the fitness costs for Cas9_gRNAs homozygous males (i.e. mate choice) and females (i.e. fecundity) differ in our phenotypic assay environment. Additionally, in contrast to our cage experiments where a model of viability-based fitness effect best matched the data (*Table 2*), we did not observe reduced viability for Cas9_gRNAs carrying flies in the individual assay (*Figure 4C*). However, the difference in male- and female-specific fitness costs in the phenotypic assay as well as the lack of difference in viability between EGFP and Cas9_gRNAs carrying flies could be explained by the limited sample size of the phenotypic assays or the assay environment itself. All phenotypic assays were based on single fly crosses conducted in vials, an environment where larvae, as well as adults, experience much less resource competition than in the densely populated cage populations, which can significantly influence the relative viability of different genotypes (*Moya et al., 1988*). Indeed, individuals with a genotype that showed reduced fecundity or mating success (but no reduced viability) in single-cross assays may not have survived to the adult stage in the more competitive cage environment, which should represent a viability cost in that system. A longer development time of individuals carrying the Cas9_gRNAs construct would also have appeared as a viability cost in our cage study but not in these fitness assays. In addition, the viability assay examined only Cas9_gRNAs/EGFP heterozygotes, which may not have suffered from off-target effects to the full extent because they received uncut 'wild-type' off-target sites from one parent that did not carry the construct.

## Evaluation of computationally predicted off-target sites

In some instances, off-target sites can be computationally predicted based on sequence similarity to gRNAs using programs such as CRISPR Optimal Target Finder (*Gratz et al., 2014*). We employed this approach to test whether we can directly observe cuts at predicted off-target sites for our Cas9_gRNAs construct. Using maximum stringency criteria to increase sensitivity, CRISPR Optimal Target Finder predicted two putative off-target sites for each of the first three gRNAs of this construct and no off-target site for its last gRNA (*Supplementary file 2*). These represent perhaps the most likely off-target sites in the genome (*Li et al., 2019*; *Pan et al., 2022*). To screen for the presence of mutations indicative of off-target cutting, we extracted genomic DNA from flies that had been homozygous for the Cas9_gRNAs construct for approximately 60 generations and performed Sanger sequencing at these six predicted off-target sites. We reasoned that if off-target cleavage at any of these six sites is responsible for the observed fitness cost (and thus occurs frequently enough to affect Cas9_gRNAs construct dynamics over less than 20 generations of our cage experiments), then we would expect to observe at least some mutations in Cas9_gRNAs homozygotes after 60 generations. However, we found only wild-type sequences with no detectable mosaicism at these sites. While direct observation of mutated sequences at this small set of computationally predicted sites would have confirmed that off-target sequence cleavage was occurring, we will argue below that off-target cutting at other genomic sites can still be the primary driver of fitness costs for the Cas9_gRNAs construct.

## Cas9HF1 homing drive

Our observation that the Cas9_gRNAs construct yet not Cas9HF1-gRNAs incurs a detectable fitness cost raises the question of whether the Cas9HF1 endonuclease can constitute a superior choice for gene drive strategies as compared to standard Cas9. As a proof of principle that Cas9HF1 is indeed a feasible alternative, we assessed two homing drives that were identical except that one used standard Cas9 while the other used Cas9HF1. The first of these drives was originally assessed in a previous study (*Champer et al., 2020c*). Both drives contain the DsRed fluorescence marker and are capable of homing by targeting EGFP with a single gRNA. Note that these drives use the same germline *nanos* promoter for expressing Cas9 as three of our previously described constructs in this study. Both drives were also placed at the same genomic site as the four constructs used in our population cage experiments and targeted the same EGFP allele that was used as a control in the cage experiments.

We first crossed male flies carrying one of the two homing drives to females with the same EGFP target site used in our cage experiments. Individuals heterozygous for the homing drive and an EGFP allele were then further crossed to flies homozygous for EGFP, or to $w^{1118}$ females for several of the

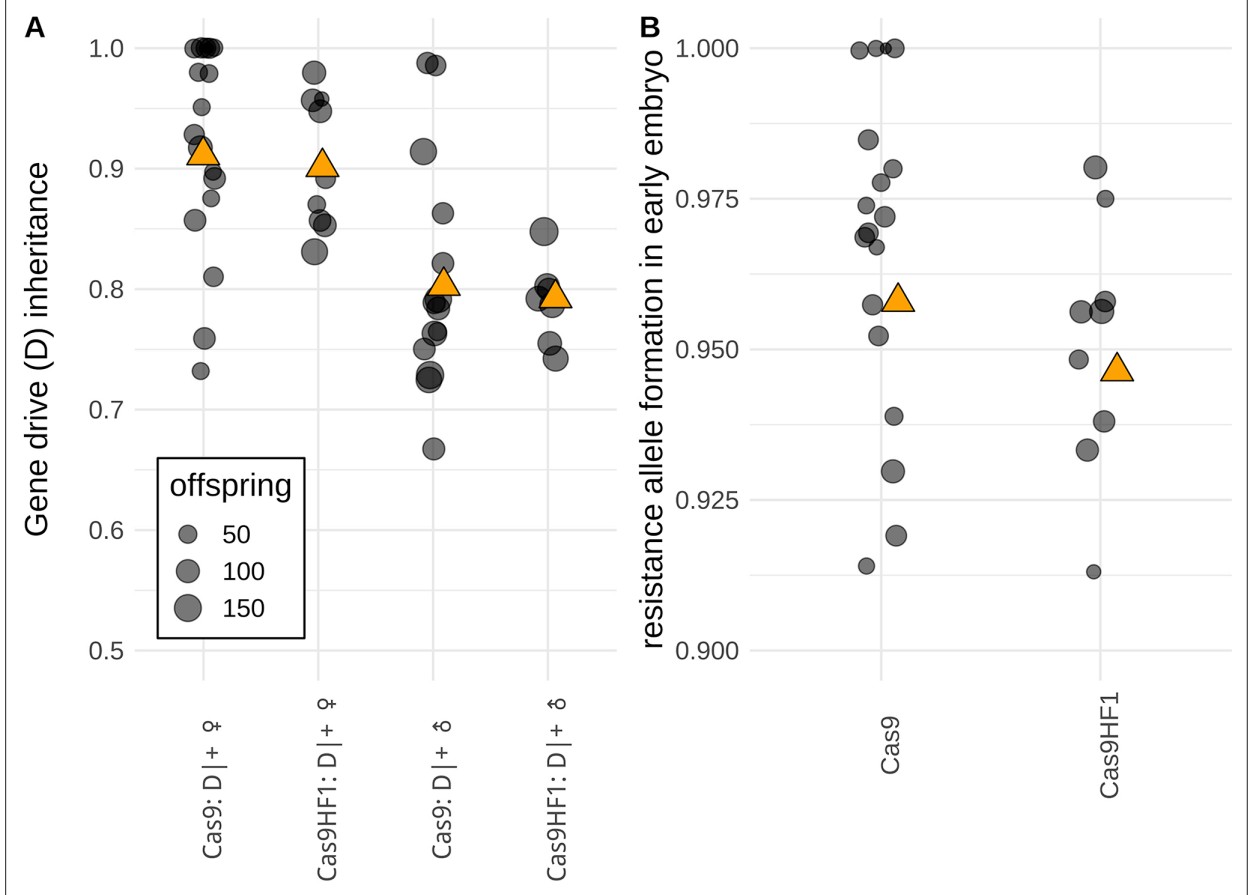

**Figure 5.** Comparison of drive performance between standard Cas9 and Cas9HF1. (**A**) The gene drive inheritance rate is defined as the proportion of offspring with DsRed fluorescence from single crosses between drive heterozygotes (D|+) and EGFP homozygotes (**B**) Resistance allele formation occurs in early embryos due to maternally deposited Cas9 and gRNAs. The resistance allele formation rate is defined as the proportion of drive heterozygous offspring that have a nonfunctional EGFP alleles (which were originally unmutated EGFP alleles inherited from the father). Each dot represents data based on a single cross. Orange triangles show the average value from all offspring combined. An alternate analysis taking potential batch effects into account led to qualitatively similar results and can be found in Data Set S1 and S2 in **Supplementary file 5**.

male drive heterozygotes. The progeny of these crosses was phenotyped for DsRed, indicating the presence of a drive allele, and EGFP, which would usually indicate the presence of an intact target allele (or more rarely, a resistance allele that preserved the function of EGFP, though we did not attempt to distinguish these in our study). Disrupted EGFP alleles that did not show green fluorescence indicated the presence of a resistance allele.

We observed similar performance between the Cas9HF1 drive and the standard Cas9 drive (*Figure 5*). Based on the overall gene drive inheritance rates (*Figure 5A*), we calculated drive conversion efficiency (i.e. the rate at which non-drive alleles were converted to drive alleles). For the Cas9HF1 drive, the drive conversion rate was estimated at 80 ± 2% for females and 59 ± 3% for males, which was not significantly different from the rates for the standard Cas9 drive (83 ± 2% for females and 61 ± 2% for males; p=0.321 for female heterozygotes and p=0.5513 for male heterozygotes, Fisher's exact test, Data Set S1 & S2 in *Supplementary file 5*). For both drives, all EGFP alleles in male heterozygotes that had not been converted to drive alleles were converted to resistance alleles, as indicated by the lack of EGFP phenotype in all progeny from crosses with $w^{1118}$ females. Both drives also had similar rates of resistance allele formation in the early embryo due to maternally deposited Cas9 (*Figure 5B*, 95 ± 1% for alleles that disrupt EGFP for Cas9HF1, and 96 ± 1% for standard Cas9, p=0.3956, Fisher's exact test, Data Set S1& S2 in *Supplementary file 5*). Together, these data demonstrate that homing drives with Cas9HF1 are capable of similar performance as drives using standard Cas9.

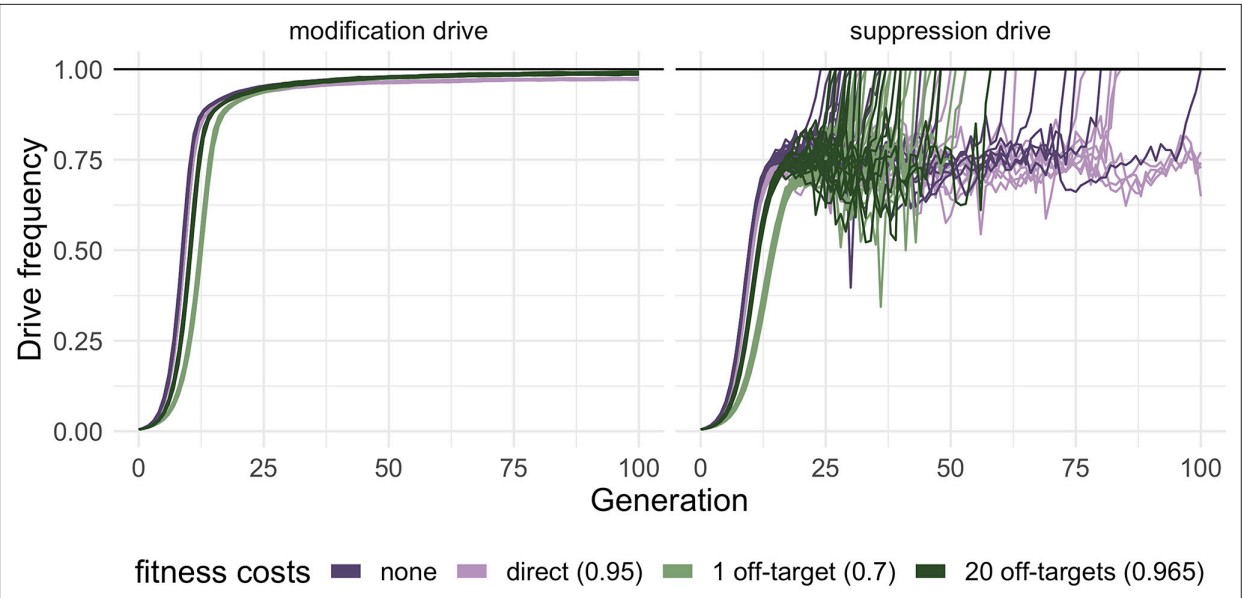

**Figure 6.** Effect of different types of fitness costs on homing drives performance. Drive allele frequency trajectories in simulated release scenarios for modification (left panel) and suppression (right panel) homing drives suffering from various types of fitness costs were plotted after an initial release of 1% heterozygous gene drive carriers. The fitness values represent the fitness of individuals homozygous for all alleles inducing a fitness cost (reference value = 1, see also Materials and methods for further details on the fitness costs of 20 off-target sites). Fitness costs were modeled as reduced viability and are multiplicative per allele. Each line represents one individual simulation (n=20 for each drive). A suppression drive frequency of 100% (black horizontal line, right panel) indicates a successful suppression of the population. For each simulation, the carrying capacity was set to 200,000, and the low-density growth rate was 10 (*Champer et al., 2020c*).

## Effect of off-target fitness costs on gene drive performance

To examine the effect of off-target fitness costs on the performance of homing drives, we extended a previously developed agent-based simulation framework for homing drives (*Champer et al., 2020c*) to include fitness costs due to off-target effects (see Methods). We used this simulation framework to investigate the invasion dynamics of both modification and suppression drives under different models of fitness costs. Specifically, we wanted to compare models with no additional fitness costs (beyond those inherent to the drive mechanism, such as when a suppression drive targets a fertility gene), models with only direct additional fitness affecting viability, and models where additional fitness costs are due to off-target cutting at a specific number of off-target sites, with cut alleles at those sites reducing viability.

Our simulated modification drive targets an essential but haplosufficient gene while also containing a rescue version of the target gene. Thus, only individuals homozygous for nonfunctional resistance alleles are nonviable. A model of this drive with only direct additional fitness costs did not fix over the course of 100 generations but approached a high equilibrium frequency due to the masking of nonfunctional resistance alleles in heterozygotes (*Figure 6*). In a model where additional fitness costs were due to off-target cuts (either at a single one or at 20 potential off-target sites), the drives proceeded toward a 100% equilibrium frequency (not quite reaching it over the course of the simulations), but they increased at a slower rate than modification drives not suffering from off-target costs, particularly for the drive with a single off-target site.

Our simulated suppression drive targets a gene that is essential but haplosufficient for female fertility without carrying a rescue allele. Thus, females with any combination of drive and nonfunctional resistance alleles are 100% sterile. We modeled the drive to be imperfect with a drive conversion rate of only 80%. As a result, it did not continually increase to fixation (*Burt, 2003*). Instead, we observed that when this drive did not incur any additional fitness costs (beyond the sterility of homozygous females), it reached an equilibrium frequency of roughly 83% before stochastic fluctuations caused by small population sizes eventually eliminated the population in all 20 simulations after 100 generations (*Figure 6*). The equilibrium frequency for a suppression drive with additional direct fitness costs tended to be lower (81%), and population suppression was not always successful over the time course

of 100 generations. Interestingly, while the initial rate of increase was slower for suppression drives with off-target fitness costs, population suppression tended to occur faster for these drives, especially when we simulated 20 off-target sites. This effect was likely because of a reduction in the average population fitness after the drive reached equilibrium frequency due to the accumulation of many detrimental cut off-target alleles, which should also lower the fitness of individuals with wild-type alleles at the drive locus.

## Discussion

Negative selection will tend to displace any alleles from the population that are sufficiently deleterious. This effect can be quantified by an allele's fitness (i.e. the relative reproductive success between carriers and noncarriers of the allele). In this study, we measured the fitness of transgenic Cas9/gRNA alleles in *D. melanogaster*, which constitute an essential component of CRISPR gene drives. A quantitative estimate of the fitness costs imposed by such constructs is critical for predicting the performance and potential limitations of this emerging technology.

Our constructs were designed to mimic a gene drive, yet without homing or any other mechanism that would facilitate super-Mendelian inheritance. This allowed us to estimate their 'baseline' fitness costs by tracking allele frequencies in cage populations. Using a maximum likelihood inference approach (*Liu et al., 2019*), we did not detect fitness costs due to Cas9/gRNA integration, expression, and on-target activity in our experiments (which together we refer to as 'direct' costs). However, we inferred a moderate fitness cost possibly resulting from off-target effects. This interpretation was further corroborated by several additional results: first, fitness costs were only detected for the construct that expressed both Cas9 and gRNAs, but not for the constructs that lacked gRNAs, suggesting that these fitness costs are primarily due to cleavage activity, rather than just the expression of Cas9 or its genomic integration. Furthermore, we did not detect any fitness costs for the construct in which Cas9 was replaced with Cas9HF1, a high-fidelity version of Cas9 designed to minimize off-target cleavage (*Kleinstiver et al., 2016*). Finally, a model where fitness costs are primarily due to off-target cuts is also consistent with the heterogeneity we observed in the frequency trajectories of the Cas9_gRNAs construct in our cage experiments, as this could be explained by the accumulation of cut alleles at the off-target sites over the course of the experiment. It would be more difficult to reconcile this observation with models where direct fitness costs are the driving factor or where fitness costs would be due to the specific genetic background of the construct flies or the health or development-related effects of the initially released flies. While the effects of off-target cutting in cells transiently exposed to Cas9 have already been extensively studied (*Kimberland et al., 2018*; *Palermo et al., 2019*; *Klein et al., 2018*; *Carroll, 2019*), our results suggest that such cutting may have more substantial negative consequences over multiple generations when the cells are continuously expressing Cas9 from a genomic source.

By inferring fitness costs from allele frequency trajectories in a population, our approach is complementary to previous studies that sought to directly detect individual off-target mutations and predict their potential phenotypic effects, including a recent study on mosquito gene drives (*Garrood et al., 2021*). Our approach, by contrast, infers the overall impact that a Cas9 allele has on reproductive success across multiple generations, thereby integrating selective effects over all life stages and affected phenotypes. Such a direct measurement of the 'fitness' of the given construct avoids many of the complexities involved in seeking to identify all off-target mutations. For example, this would likely require time-resolved whole-genome sequencing on the population level so that also rare but potentially consequential mutations can be detected, which could be anywhere in the genome. Even when all such mutations can be accurately identified, predicting their potential effects on fitness will not always be straightforward.

These complexities are further underlined by the fact that we did not actually detect mutated sequences at any of the top six computationally predicted off-target sites for our Cas9_gRNAs construct. One potential explanation is that all these predicted sites in fact contain several nucleotide mismatches (often in the critical Protospacer Adjacent Motif (PAM)-proximal 12 nucleotides), which should make cleavage less likely. These predictions by CRISPR Optimal Target Finder are based on *in vivo* cleavage specificity in cell lines where off-target cutting is known to occur more frequently than in animals (*Gratz et al., 2014*). However, our system may be substantially different because it involves

strong *in vivo* germline and early embryo expression across several generations, and it is unclear whether that would lead to higher or lower expected cleavage rates than in cell line studies.

At first glance, the lack of mutated alleles at this small set of computationally predicted sites may seem to contradict the interpretation that off-target cleavage is the primary driver of fitness costs for the Cas9_gRNAs construct. However, we believe it is likely that off-target cleavage did still occur, yet at other genomic sites than those we tested. Indeed, off-target cleavage can frequently occur at sites that are difficult to predict computationally (*Li et al., 2019*; *Höijer et al., 2020*), and it has been observed at sites with 5 or more mismatches (*Pan et al., 2022*; *Tsai et al., 2015*; *Fu et al., 2013*; *Chaudhari et al., 2020*), meaning there are possibly thousands of potential off-target sites in the fly genome. While the individual cleavage rates at most of these sites might be vanishingly low, the combined genome-wide rate could still be substantial. Off-target sites could also simply fail to be predicted if they occur in regions of the genome that are poorly assembled. Importantly, even rare off-target cleavage events can still have a strong impact on fitness. For example, if deleterious cleavage events occur in only 10% of drive carriers but cause them to have a 50% reduction in viability or fertility, this would still amount to a net fitness cost of 5% for the construct. Such events do not need to occur consistently at the same sites to be able to substantially alter construct frequency trajectories in aggregate. For all these reasons, we believe that our inability to detect mutated alleles at a small set of computationally predicted off-target sites does not rule out the presence of off-target cleavage at other sites. Indeed, a key benefit of our approach is that it allows us to estimate fitness costs without having to rely on the detection of specific off-target cleavage events and subsequent evaluation of their direct potential effects on fitness, which can be complicated.

Nevertheless, our study does have some important limitations. The maximum likelihood framework for fitness estimation necessarily relied on a simplistic ranking based on AICc values (a goodness-of-fit measure) of five highly idealized selection models. Although AICc values are useful for choosing the model with the best fit, we must not systematically dismiss alternative evolutionary scenarios. Thus, while our model comparison points to off-target fitness costs affecting viability as the most plausible explanation for the observed construct frequency dynamics among the models we tested, the suggested presence of off-target fitness costs in our experiment does not rule out other factors contributing to fitness costs of CRISPR endonucleases in *Drosophila*.

Another simplified feature of our model is that we only used a single genomic site with codominant fitness costs to represent off-target cutting. In reality, there could be many possible off-target sites, with variable types of alleles produced after cleavage, different fitness costs, dominance relationships, degrees of genetic linkage, and possibly even epistatic interactions between them; not to mention the plethora of alternative factors that could contribute to the observed fitness costs of CRISPR endonucleases in our study. For example, a detrimental mutation could have occurred, by chance, in only a subset of our transgenic fly lines carrying the Cas9_gRNAs construct. While our 'initial off-target' model—which assumes that the construct insertion site and the idealized off-target site are unlinked—did not fit the Cas9_gRNAs construct dynamics well, a model accounting for intermediate linkage might have resulted in a better fit. Given the limited number of data points in our cage experiments, together with the large number of conceivable models, we cannot rule out that more complex selection scenarios could ultimately provide a better fit to our data.

The same holds true for the inference of different fitness components. While we did compare models where fitness affected viability versus models where it affected fecundity and mate choice, we believe that any conclusions from these comparisons should be taken with a grain of salt, given how many assumptions still went into each model (e.g. multiplicative fitness costs, same costs in males and females, and equal costs for fecundity and mate choice). Indeed, while our maximum likelihood analysis of the cage experiments ranked the viability model higher than the fecundity/mate choice model, we did observe reduced fecundity and mating success of genotypes carrying the Cas9_gRNAs construct in our phenotypic assays, while not finding a substantial effect on viability. As explained above, this could be due to a lower power of the phenotypic assays caused either by limited sample size, or the fact that only heterozygotes were studied in the viability assay. Furthermore, it is possible that many of those individuals with reduced fecundity and mating success in the phenotypic assays would not have survived into adult stage in the cage populations due to increased larval competition at higher densities. In summary, we hypothesize that environmental-dependent fitness effects (*Yang et al., 2022*) and the dependencies between reduced fecundity/mate choice and viability could

explain the discrepancies between our fitness estimates in complex (cage) and simple (vial) environments and should be investigated in future studies.

Though considerable uncertainty remains regarding the precise nature of the fitness effects of our constructs, the overall finding that our Cas9_gRNAs construct imposes a moderate fitness cost is robust. It remains to be seen how well these results translate to other applications that involve genomic integration of Cas9 endonucleases. If these fitness cost indeed stem from off-target cleavage, they could vary substantially depending on the specific target sequence(s), genome composition, and expression patterns of Cas9 and gRNAs (*Zhang et al., 2015*; *Gisler et al., 2019*). While we specifically selected gRNAs with a low number of predicted off-target sites to minimize such effects, this may not be possible for every application. Some applications may also require the use of different promoters with higher somatic expression rates than *nanos* (*Gantz et al., 2015*; *Hammond et al., 2016*; *Champer et al., 2017*; *Champer et al., 2018*), which could increase fitness costs caused by off-target cleavage. On the other hand, Cas9 expression may be lower in some organisms or at other genomic sites, and certain applications might require fewer than the four gRNAs we included in our constructs, thereby potentially reducing off-target effects.

Our results have important implications for the modeling of gene drive approaches. Thus far, only direct fitness costs have been included in such studies, arising from the CRISPR nuclease itself, a payload gene, or cleavage of the intended target site. It is well known that such direct fitness costs can reduce the power of a suppression drive (*Champer et al., 2021b*; *Deredec et al., 2008*; *North et al., 2020*) and reduce the persistence of a modification drive in the face of resistance alleles (*Unckless et al., 2017*). If these costs are in fact lower than 5%, as suggested by our study in *Drosophila* with the *nanos* promoter, then they would not be expected to substantially impede the spread of suppression drives, though they could still suffer from other forms of direct fitness costs such as haploinsufficiency of the target gene and somatic Cas9 expression and cleavage, the latter of which has proven to be a particular determinate to some drives in mosquitoes (*Hammond et al., 2016*). The direct fitness of modification drives would be largely determined by their cargo gene(s) and possibly their rescue efficiency if they involve use of a recoded gene (*Champer et al., 2020b*; *Oberhofer et al., 2019*; *Champer et al., 2020d*; *Adolfi et al., 2020*).

On the other hand, if off-target effects are indeed the primary cause of fitness costs of a drive with otherwise low direct fitness costs, this should result in different population dynamics. Our modeling indicates that for a modification drive, such off-target effects should only slow the drive initially. After the drive has spread through most of the population and cleaved off-target alleles had time to accumulate, resistance alleles would not be as selectively advantageous. Thus, it would take them longer to outcompete drive alleles in the long run as compared to a scenario with primarily direct fitness costs. A suppression drive still suffers from cuts at off-target sites in a manner more closely resembling direct fitness costs, because these effects come into play during the early spread of a drive, often the most critical period in determining the fate of a suppression drive (*Champer et al., 2021b*; *Deredec et al., 2008*; *North et al., 2020*). However, if the rate at which off-target mutations arise is sufficiently low, then mutated off-target sequences may not have a large effect on population dynamics. This has been shown in a recent study on a suppression drive with a germline-restricted promoter and a single gRNA that successfully eliminated a mosquito cage population before substantial amounts of off-target cleavage could occur (*Garrood et al., 2021*). Indeed, our results suggest that off-target fitness costs could potentially support a suppression drive if off-target sites are not genetically linked to the drive locus. In that case, cut off-target alleles will increasingly be found also in individuals that do not carry the drive, reducing their ability to persist in the face of an imperfect drive. In complex, spatially explicit scenarios, such drives may still be weaker, since any long-term benefits may be rendered void by their reduced rate of spread, which should make them more vulnerable to the chasing effect (*Champer et al., 2021b*). Though we did not model it explicitly, threshold-based drives would likely suffer more from off-target fitness costs due to the critical importance of fitness of the released individuals. Specifically, off-target fitness costs should increase the necessary introduction threshold for these drives.

We demonstrated that Cas9HF1, which largely eliminates off-target cleavage (*Kleinstiver et al., 2016*), does not induce substantial negative fitness effects in our cage populations when used as a replacement for standard Cas9. Furthermore, we showed that homing drives with either form of Cas9 perform similarly for at least one gRNA target site. We, therefore, recommend that gene drives, as

well as other applications that require the genomic integration of CRISPR endonucleases, should consider moving from standard *Streptococcus pyogenes* Cas9 to higher fidelity versions that can effectively minimize off-target effects. This would have the added advantage of reducing the generation of unanticipated genetic changes in natural populations from off-target cleavage and repair. One potential drawback of these nucleases is that they tend to have a lower cleavage rate that can depend on the specific gRNA sequence employed (*Casini et al., 2018*; *Tan et al., 2019*; *Chatterjee et al., 2020*; *Xie et al., 2020*; *Slaymaker et al., 2016*; *Lee et al., 2019*). In practice, this may reduce the number of available gRNA target sites and increase the need for the initial evaluation of gRNA targets. Newer improved forms of Cas9 (*Casini et al., 2018*; *Tan et al., 2019*; *Xie et al., 2020*; *Slaymaker et al., 2016*; *Lee et al., 2019*), including ones with an expanded range of target sites (*Chatterjee et al., 2020*), promise to ameliorate this issue.

In conclusion, we have demonstrated that genomic CRISPR/Cas9 expression in *D. melanogaster* can impose a moderate level of fitness costs, presumably via off-target effects. Our results further indicate that these fitness costs can be minimized by using a high-fidelity endonuclease with reduced off-target cleavage. Future studies should investigate whether our conclusions hold in other experiments involving different constructs, target sites, and organisms.

## Materials and methods
### Plasmid construction
The starting plasmid pDsRed (Addgene plasmid #51019) was provided by Melissa Harrison, Kate O'Connor-Giles, and Jill Wildonger, pnos-Cas9-nos (*Port et al., 2014*; Addgene plasmid #62208) was provided by Simon Bullock, and VP12 (*Kleinstiver et al., 2016*; Addgene plasmid #72247) was provided by Simon Bullock. Starting plasmids ATSacG, TTTgRNAtRNAi, TTTgRNAt, BHDgN1c, and BHDgN1cv3 were constructed in a previous study (*Champer et al., 2020c*). Restriction enzymes for plasmid digestion, Q5 Hot Start DNA Polymerase for PCR, and Assembly Master Mix for Gibson assembly were acquired from New England Biolabs. Oligonucleotides and gBlocks were obtained from Integrated DNA Technologies. JM109 competent cells and ZymoPure Midiprep kit from Zymo Research were used to transform and purify plasmids. Cas9 gRNA target sequences were identified by the use of CRISPR Optimal Target Finder (*Gratz et al., 2014*). A list of DNA fragments, plasmids, primers, and restriction enzymes used for cloning each construct can be found in *Supplementary file 3*. The annotated sequences of the final construct insertion plasmids can be found in *Supplementary file 4* (ApE format, http://biologylabs.utah.edu/jorgensen/wayned/ape).

### Generation of transgenic lines
Injections were conducted by Rainbow Transgenic Flies. The donor plasmid (Cas9_gRNAs, Cas9_no-gRNAs, no-Cas9_no-gRNAs, Cas9HF1_gRNAs, or BHDgNf1v2; ~500 ng/µL) was injected along with plasmid BHDgg1c (or TTTgU1 for BHDgNf1v2; *Champer et al., 2020c*; ~100 ng/µL), which provided additional gRNAs for transformation, and pBS-Hsp70-Cas9 (~500 ng/µL, from Melissa Harrison, Kate O'Connor-Giles, and Jill Wildonger; Addgene plasmid #45945) providing Cas9. A 10 mM Tris-HCl, 100 µM EDTA solution at pH 8.5 was used for the injection. Most constructs were injected into $w^{1118}$ flies, but BHDgNf1v2 was injected into flies with ATSacG (*Champer et al., 2020c*). Transformants were identified by the presence of DsRed fluorescent protein in the eyes, which usually indicated successful construct insertion. Correct insertion of the lines was confirmed by sequencing. Independently obtained lines were used for each population cage, except for the Cas9_gRNAs construct, where a total of four lines were used in the seven population cages, and replicates 1–4 were founded with the same line.

### Maintenance of transgenic flies with gene drives
To minimize the risk of accidental release, all flies with an active homing gene drive system were kept at the Sarkaria Arthropod Research Laboratory at Cornell University under Arthropod Containment Level 2 protocols in accordance with USDA APHIS standards. In addition, the synthetic-target-site drive system (*Champer et al., 2019a*), used in both gene drive lines tested here, prevents drive conversion in wild-type flies, which lack the EGFP target site. All safety standards were approved by the Cornell University Institutional Biosafety Committee.

## Experimental fly populations

The experimental fly populations were maintained on Bloomington Standard medium in 30 × 30 × 30 cm$^3$ fly cages (Bugdorm). Flies were kept at constant temperature (25°C, 14 hr light, 10 hr dark), with non-overlapping generations. Zero-day-old to two-day-old flies of one generation were allowed to lay eggs on fresh medium (eight food bottles per cage) for 24 hours. Population sizes were controlled via this limited egg-lay period, which led to fluctuations in the number of flies per generation (*Figure 2— figure supplement 1*). Some experiments experienced bottlenecks due to high variation in food moisture content (resulting in either high or low larvae density). After the egg lay, the adults were frozen at –20°C for later phenotyping, and the new generation was allowed to develop for 11–12 days before a fresh medium was provided and a new generation cycle started.

The ancestral generation of each cage was generated by allowing homozygous EGFP flies and flies homozygous for the construct to deposit eggs for 24 hours separately from each other in four food bottles each. These eight egg-containing bottles were put in the fly cages to start one experimental fly population. All experiments started by crossing the construct and EGFP homozygotes as described above, except for replicates 1 and 2 of Cas9_gRNAs. These two experimental populations were set up with all three genotypes that originated from the same batch that included heterozygotes and both homozygotes. While construct homozygotes of Cas9_no-gRNAs, no-Cas9_no-gRNAs, and Cas9HF1_gRNAs were of the same age as the EGFP homozygotes they were mixed with to start the experiments, the age differed between EGFP and construct homozygotes for Cas9_gRNAs replicate 1, 2, 5, 6, and 7. To avoid confounding maternal effects on the construct frequency dynamics, we excluded for each of these replicates the first generation from the analysis. The full data set including the removed time points can be found in *Supplementary file 5*. Seven replicates of Cas9_gRNAs, and two replicates each for Cas9_no-gRNAs, no-Cas9_no-gRNAs, and Cas9HF1_gRNAs were maintained.

## Phenotyping experimental fly populations

The dominant fluorescent markers (EGFP and DsRed) allow a direct readout of the genotype by screening the fluorescent phenotype of an individual fly. Flies that are only red fluorescent are construct homozygotes, flies that are only green fluorescent do not carry any construct, and flies that are fluorescent for both colors carry one construct copy.

For each experimental population and generation, all individuals were screened for their genotypes using either a stereo dissecting microscope in combination with the NIGHTSEA system or an automated image-based screening pipeline we specifically developed for this purpose. Quantifying phenotypic traits (e.g. pupae size, the amount of laid eggs) in an automated way has been done successfully before in *Drosophila* (*Reeves and Tautz, 2017*; *Nouhaud et al., 2018*). In our image-based screening pipeline, three pictures were taken for each batch of flies: a white light picture to determine the number and the position of the flies, one fluorescent picture filtered to screen for DsRed, and one fluorescent picture filtered to screen for EGFP expression.

We used a Canon EOS Rebel T6 with an 18–55 mm lens for image acquisition. The camera was held in a fixed position by a bracket 25 cm above the frozen flies spread on a black poster board. NIGHTSEA light heads (Green and Royalblue) were used as light sources. The light sources both white and fluorescent light were covered with paper tissue for diffusion. For the fluorescent pictures, barrier filters (Tiffen 58 mm Dark Red #29; Tiffen 58 mm Green #58) were used, attached with a magnetic XUME Lens/Filter system to the camera. Except for the filter change, the camera was fully controlled through a PC interface (EOS Utility 2 software). The focus was set automatically under white light and was kept constant for the fluorescent pictures. First, a white light picture (F 5.6, ISO 100, exposure time 1") was taken to determine the number and positions of the flies. Second, a picture under NIGHTSEA Green light with the Tiffen Dark Red #29 filter (F 5.6, ISO 400, exposure time 30") was taken to determine, whether flies express DsRed. Third, a picture under NIGHTSEA Royal Blue with the Tiffen Dark Green #58 barrier filter (F 5.6, ISO 400, exposure time 25") was taken to screen flies for EGFP expression.

We used the ImageJ distribution Fiji (v 2.0.0-rc-69/1.52 p) (*Schindelin et al., 2015*; *Schindelin et al., 2012*) to process and analyze the picture sets with an in-house ImageJ macro: the three multi-channel images were split into the respective red, green, and blue image components. Further analysis included the red and the green image component of the white light picture, the red image component of the red fluorescent picture, and the green image component of the green fluorescent picture.

The four remaining images were merged into a stack, and we performed slice alignment (matching method: normalized correlation coefficient) based on a selected landmark using the plugin Template_ Matching.jar (*Tseng, 2018*). We used a rectangular piece of white tape on the black poster board as a landmark. To obtain the contours of the flies, we calculated the difference between the red and the green image component of the white light picture and applied a median and a Gaussian filter (radius = 3 pixels). After that, the picture was binarized using global thresholding (option: Max Entropy; *Kapur et al., 1985*). The binary image was post-processed (functions: Fill Holes, Open) before the position, and the size of individual particles (= flies) were determined with the Analyze Particles method of ImageJ (minimum size = 750 pixels$^2$). To account for translocations that have not been corrected for by the slice alignment (e.g. when the position of the fly changed slightly), the convex hull for each particle was calculated and enlarged by 20 pixels. A median filter (radius = 2 pixels) was applied to both fluorescent pictures before each particle (= fly) was scanned by a human investigator for the eye fluorescent pattern in both fluorescent pictures. We compared the image-based screening pipeline to the screening method using a stereo dissecting microscope and found that the estimated genotype frequencies deviate not more than 1% from each other (n=646 flies, 4 picture sets).

## Phenotypic assays

We measured three fitness proxies for flies carrying Cas9_gRNAs constructs: mate choice, fecundity, and viability. As in the large cage populations, the dominant fluorescent markers allowed us to infer the genotype of an individual fly by phenotyping it for its eye color. All phenotypic assays were conducted on Bloomington Standard medium and under the same temperature (25°C) and light conditions (14 hr light/10 hr dark) as the caged populations. The statistical analysis of the phenotypic assays was conducted in R (v3.6.0) (*R Development Core Team, 2019*).

### Mate choice

We conducted a mate choice assay to test for mating preferences of EGFP homozygous females. Individual 2-day-old virgin EGFP homozygous females were set up with one EGFP homozygous male and one Cas9_gRNAs homozygous male of the same age in a vial. After 24 hours, the adult flies were removed, and the genotypes of the eclosed offspring were assessed after 11–12 days. If the EGFP homozygous female has mated only with the male of the same genotype, only homozygous offspring is expected. We tested for deviations from an expected equal frequency of offspring genotypes under the null hypothesis of no mate preference via a binomial test.

### Fecundity

We assessed the fecundity of EGFP homozygous, heterozygous, and Cas9_gRNAs homozygous females in individual crosses with EGFP homozygous males. Each individual single 2-day-old virgin female of a distinct genotype was crossed with one EGFP homozygous male of the same age. Crosses were flipped on fresh medium every 24 hours, and eggs were counted manually using a stereo dissecting microscope. Fecundity was defined as the total number of laid eggs per female over three consecutive days. To assess the impact of female genotype we fitted a linear model using function lm() with fecundity as a response. The female genotype was the only fixed effect in the model. The residuals were both normally distributed and showed variance homogeneity, meeting all assumptions of a linear model. None of the used model diagnostics (Cook's distance, DFbetas, leverage *Fox and Monette, 1992* calculated with the R package car [v3.0–3] *Fox and Weisberg, 2019*) indicated strongly influential cases or outliers. We used the R package emmeans (v1.4.7) (*Lenth, 2020*) to conduct pairwise comparisons of the three assessed female genotypes.

### Viability

We measured viability as the fraction of heterozygous offspring out of the total number of offspring of single crosses between EGFP homozygous males and heterozygous females. Single 2-day-old heterozygous virgin females were crossed each with one EGFP homozygous male of the same age. After 24 hours, the adult flies were removed, and the genotypes of the eclosed offspring was assessed after 11–12 days. If the genotype does not influence viability, we expect 50% of the offspring to be heterozygotes. Fraction of heterozygous offspring was tested for normality with an Anderson-Darling

test (function ad.test() in the R package nortest [v1.0–4] *Gross and Ligges, 2015*). We then used a one-sample t-test against a population mean of 0.5 for the fraction of heterozygous offspring.

## Phenotype data analysis, Cas9HF1 homing gene drive

For each individual cross, all offspring was screened for their respective genotypes using a stereo dissecting microscope in combination with the NIGHTSEA system. When calculating drive parameters, we pooled offspring from the same type of cross together and calculated rates from the combined counts. A potential issue of this pooling approach is that batch effects could distort rate and error estimates (offspring were raised in separate vials with different parents). To account for such effects, we performed an alternate analysis as in previous studies *Champer et al., 2020b*; *Champer et al., 2020c* by fitting a generalized linear mixed-effects model with a binomial distribution using the function glmer and a binomial link function (fit by maximum likelihood, Adaptive Gauss-Hermite Quadrature, nAGQ = 25). This allows for variance between batches, usually resulting in different rate estimates and increased error estimates. Offspring from a single vial was considered as a distinct batch. This analysis was performed using the R statistical computing environment (v3.6.1) (*R Development Core Team, 2019*) with packages lme4 (1.1–21, https://cran.r-project.org/web/packages/lme4/index.html) and emmeans (1.4.2, https://cran.r-project.org/web/packages/emmeans/index.html). The R script we used for this analysis is available on GitHub (https://github.com/MesserLab/Binomial-Analysis; *Champer, 2019*). The results were similar to the pooled analysis and are provided in *Supplementary file 5* (Data Sets S1–S2).

## Genotyping

We used a PCR-based genotyping approach to confirm mutated gRNA target sites (i.e. active gRNA) in the progeny of individuals carrying either the Cas9_gRNAs or the Cas9HF1_gRNAs construct. For this, flies were frozen, and DNA was extracted by grinding in 30 µL of 10 mM Tris-HCl pH 8, 1 mM EDTA, 25 mM NaCl, and 200 µg/mL recombinant proteinase K (ThermoScientific) followed by incubation at 37°C for 30 min and then 95°C for 5 min. The DNA was used as a template for PCR using Q5 Hot Start DNA Polymerase from New England Biolabs. The region of interest containing gRNA target sites was amplified using DNA oligo primers AutoDLeft_S2_F and AutoDRight_S2_R. PCR products were purified after gel electrophoresis using a gel extraction kit (Zymo Research). Purified products were Sanger sequenced and analyzed with ApE (http://biologylabs.utah.edu/jorgensen/wayned/ape). The location of the construct was similarly confirmed using PCR with primers AutoC_S2_F and EGFPaLeft_S_R.

## Off-target site assessment

Potential off-target sites for each of the four gRNAs in the Cas9_gRNAs construct were predicted by CRISPR Optimal Target Finder (http://targetfinder.flycrispr.neuro.brown.edu/; *Gratz et al., 2014*) using maximum stringency settings. The resulting set of predicted sites is listed in *Supplementary file 2*. Primers for amplification of each site were designed using Primer3Plus (https://dev.primer3plus.com/index.html). Genomic DNA was extracted from approximately 20–30 flies that had been homozygous for the Cas9_gRNAs construct for approximately 60 generations (as well as similar $w^{1118}$ flies as controls). Sanger sequencing was performed on PCR products.

## Maximum likelihood framework for fitness cost estimation

To estimate the fitness costs of the different transgenic constructs from time-resolved genotype frequencies in our *D. melanogaster* cage experiments, we modified a previously developed maximum likelihood inference framework (*Liu et al., 2019*). Specifically, we extended the original model to an autosomal two-locus model, where the first locus represents the construct insertion site and the second locus represents an unlinked idealized cut site. In this model, cleavage at the cut site could represent in principle the net effects of non-specific DNA modifications ('off-target' effects) as well as the effects of cleavage at the desired gRNA target site (i.e. target site activity). However, the latter is not expected to impose any fitness costs for our constructs due to the intergenic location of the target site. Thus, we refer to the idealized cut site as an 'off-target' site. At the construct locus, the two possible allele states are EGFP/construct (observed by the eye fluorescence phenotype); at the off-target site, the two possible states are uncut/cut (not directly observed). The two loci are assumed

to be autosomal and unlinked. Thus, there are nine possible genotype combinations an individual could have in our model. Unless stated otherwise, we assumed that the construct homozygotes used for the ancestral generation of a cage are cut/cut homozygotes at the idealized off-target site. Since the construct is not homing (the gRNA target sites are positioned on a non-homologous chromosome arm, *Figure 1*), the allelic state of a single individual cannot change at the construct locus. By contrast, the allelic state at the off-target locus can be altered by cutting events in the germline or in the early embryo phase. Germline cutting will only impact the genotype of offspring in the next generation, while embryo cutting will directly change the individual's genotype and hence expose it to any potential fitness effects of this new genotype. Both the germline and embryo off-target cut rates were set to 1 in our model. This means that any uncut allele at the off-target locus will be cut in the germline if the individual carries at least one construct allele (germline cut rate = 1). Furthermore, individuals always become cut/cut homozygotes if their mother carried a least one construct allele (embryo cut rate = 1; we assume that maternally deposited Cas9/gRNA is present in all such embryos). Similar to the previously developed maximum likelihood inference framework (*Liu et al., 2019*), our two-locus inference framework does not take sampling variance of genotype frequencies into account, because all adults of an experimental population were screened for their respective genotypes. The maximum likelihood inference framework builds upon the multinomial distributions, which allows us to estimate the effective population size alongside the selection parameters (*Liu et al., 2019*).

A full inference model for the potential fitness costs of construct alleles and cut off-target alleles that includes all three previously implemented types of selection (mate choice, fecundity, and viability) would feature a vast number of parameters that would be difficult to disentangle (*Liu et al., 2019*). For simplicity and to avoid overfitting, we, therefore, reduced model complexity with a series of assumptions: first, potential fitness costs were assumed to be equal for both sexes, because previous power evaluations have shown that the maximum likelihood inference framework cannot detect sex-specific differences for an autosomal locus if the anticipated fitness costs are small (*Liu et al., 2019*). Second, we either included only viability selection in the model or included only mate choice (i.e. relative mating success for males with a particular genotype, reference value = 1) and fecundity selection (i.e. relative fecundity for females with a particular genotype, reference value = 1), both of equal magnitude. We further considered all fitness effects to be multiplicative across the two loci and for the two alleles at each locus (e.g. a construct homozygote would have a fitness equal to the square of a construct/EGFP heterozygote, given the same genotype at the off-target site). This results in two much more tractable inference models (viability and fecundity/mate choice) with only three parameters overall: the effective population size ($N_e$), the relative fitness of construct/EGFP heterozygotes versus EGFP homozygotes (the 'direct fitness parameter'), and the relative fitness of cut/uncut heterozygotes versus uncut homozygotes (the 'off-target fitness parameter').

## Modeling drive performance in the presence of off-target fitness costs

We modified a previously developed agent-based simulation framework (*Champer et al., 2020c*) to examine the expected effect of off-target fitness costs on the long-term performance of modification and suppression gene drives. For this, we simulated either 1 or 20 genetically unlinked off-target sites that were all in linkage equilibrium with the drive locus. We investigated fitness scenarios mimicking the characteristics of previous *Drosophila* homing drives, but allowing for higher efficiency by reducing the embryo resistance allele formation rate (*Champer et al., 2020c*; *Yang et al., 2022*; *Champer et al., 2020d*). Specifically, we modeled a homing drive with an 80% drive conversion rate, 10% germline resistance allele formation rate, and a 5% embryo resistance allele formation rate due to maternally deposited Cas9 and gRNA. All resistance alleles were assumed to be nonfunctional due to the use of multiplexed gRNAs and/or conserved functional target sites. Because the embryo resistance allele formation rate at the target site was low (5%), we simulated cuts at off-target sites exclusively in the germline. The off-target cut rate was set to 100% for simulations with a single off-target site and to 5% per site for simulations with 20 off-target sites. The total fitness cost of the 20 off-target sites was set to give the same fitness when half their sites were cut as the single off-target site when both sites were cut. This increased fitness cost for the 20 sites partially compensates for the substantially reduced cut rate at individual sites when considering short-term drive dynamics, which would determine the parameter inference from our cage experiments. A comprehensive description of the underlying modeling framework can be found in reference *Champer et al., 2020c*.

## Availability of data and materials

The annotated sequences of the final construct insertions are available in ApE format (*Supplementary file 4*). The raw counts of each experimental population (different constructs and the Cas9/Cas9HF1 homing drives) can be found in *Supplementary file 5*. The macro of the image-based screening pipeline is available on GitHub (https://github.com/MesserLab/CRISPR-Cas9-fitness-effects, copy archived at swh:1:rev:3b4fec78e1678470fb51530b22fffd848dc7ce00, *Langmüller, 2022*), a picture sample set for the image-based screening pipeline can be found in *Supplementary file 6*. The raw data of the phenotypic assays can be found in *Supplementary file 7*. The maximum likelihood inference framework was implemented in R (v 3.6.0) (*R Development Core Team, 2019*), and is available together with all necessary scripts to reproduce the results on GitHub (https://github.com/MesserLab/CRISPR-Cas9-fitness-effects).

## Acknowledgements

This study was supported by the National Institutes of Health awards R21AI130635 to JC, AGC, and PWM, award F32AI138476 to JC, and award R01GM127418 to PWM. JC was also supported by laboratory starting funding from Peking University. AML was supported by Vetmeduni Vienna Funds and an Austrian Science Funds grant (FWF; DK W1225-B20) awarded to Christian Schlötterer and an Austrian Marshall Plan Foundation fellowship. We thank Marlies Dolezal for helpful advice on the statistical analysis of the phenotypic assays, and Charles Mazel and Guy Reeves for support in developing the image-based screening pipeline.

## Additional information

### Competing interests

Philipp W Messer: Reviewing editor, *eLife*. The other authors declare that no competing interests exist.

### Funding

| Funder | Grant reference number | Author |
| --- | --- | --- |
| National Institutes of Health | R21AI130635 | Jackson Champer<br>Philipp W Messer<br>Andrew G Clark |
| National Institutes of Health | F32AI138476 | Jackson Champer |
| National Institutes of Health | R01GM127418 | Philipp W Messer |

The funders had no role in study design, data collection and interpretation, or the decision to submit the work for publication.

### Author contributions

Anna M Langmüller, Conceptualization, Formal analysis, Funding acquisition, Investigation, Visualization, Methodology, Writing – original draft, Writing – review and editing; Jackson Champer, Conceptualization, Resources, Formal analysis, Supervision, Funding acquisition, Investigation, Visualization, Methodology, Writing – original draft, Writing – review and editing; Sandra Lapinska, Lin Xie, Matthew Metzloff, Yineng Xu, Investigation; Samuel E Champer, Jie Du, Investigation, Methodology; Jingxian Liu, Methodology; Andrew G Clark, Resources, Supervision, Funding acquisition, Writing – review and editing; Philipp W Messer, Conceptualization, Resources, Supervision, Funding acquisition, Methodology, Writing – original draft, Project administration, Writing – review and editing

### Author ORCIDs

Anna M Langmüller http://orcid.org/0000-0002-6102-8862
Jackson Champer http://orcid.org/0000-0002-3814-3774
Matthew Metzloff http://orcid.org/0000-0002-6108-5031

Samuel E Champer 🔾 http://orcid.org/0000-0002-4559-7627
Jingxian Liu 🔾 http://orcid.org/0000-0002-2172-3297
Yineng Xu 🔾 http://orcid.org/0000-0002-4473-4052
Philipp W Messer 🔾 http://orcid.org/0000-0001-8453-9377

### Ethics

All flies with an active homing gene drive system were kept at the Sarkaria Arthropod Research Laboratory at Cornell University under Arthropod Containment Level 2 protocols in accordance with USDA APHIS standards. All safety standards were approved by the Cornell University Institutional Biosafety Committee.

### Decision letter and Author response

Decision letter https://doi.org/10.7554/eLife.71809.sa1
Author response https://doi.org/10.7554/eLife.71809.sa2

## Additional files

### Supplementary files

• Supplementary file 1. Model comparison of Cas9_no-gRNAs, no-Cas9_no-gRNAs, Cas9HF1_gRNAs. All cut parameters were set to 1.

• Supplementary file 2. Predicted off-target sequences for gRNAs 1–4. Lowercase letters indicate a mismatch between the potential off-target sequence and the gRNA sequence.

• Supplementary file 3. Plasmid construction overview. For each construct, DNA fragments, plasmids, primers, and restriction enzymes used for cloning are listed.

• Supplementary file 4. Annotated sequences of the final construct insertion plasmids.

• Supplementary file 5. Raw counts of each experimental population (different constructs and the Cas9/Cas9HF1 homing drives).

• Supplementary file 6. Picture sample set for the image-based screening pipeline.

• Supplementary file 7. Raw data of the phenotypic assays.

• Transparent reporting form

### Data availability

All data generated or analyzed during this study are included in the manuscript and supporting files.

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
