## [Editor Report]

The manuscript describes an attempt to assess fitness costs of CRISPR/Cas9 endonucleases in the context of gene drive in *Drosophila melanogaster* by looking at direct fitness costs of the transgene and indirect fitness costs due to off-target cleavage. The authors performed experimental cage population studies and a maximum-likelihood approach to disentangle the contribution of direct and off-target-related fitness costs. The combined experimental and mathematical approach allows the authors to conclude that off-target cleavage is largely responsible for the observed fitness costs, although no mutated alleles were detected at the most likely computational predicted off-target sites. The authors also use a high-fidelity Cas9 nuclease (Cas9HF) to confirm reduced fitness costs probably due to increased cleavage specificity. The data are of interest for CRISPR/Cas9 applications in general and for gene drive applications in particular and the manuscript is of interest to a wide range of readers.

---

## [Decision Letter]

**Decision letter after peer review:**

Thank you for submitting your article "Fitness effects of CRISPR endonucleases in *Drosophila melanogaster* populations" for consideration by *eLife*. Your article has been reviewed by 3 peer reviewers, and the evaluation has been overseen by a Reviewing Editor and Patricia Wittkopp as the Senior Editor. The following individuals involved in review of your submission have agreed to reveal their identity: Steven Russell (Reviewer #2).

Essential revisions:

The results seem still premature and the manuscript will require substantial revisions. Please consider all the suggestions form the reviewers. Below is a list of essential revision requirements:

1) More details on how the transgenic lines were generated, and how they were tested, including sequencing verification of the integrity of the constructs.

2) More detailed explanation of the experimental designs, and a clearer presentation of the data.

3) Provide some direct evidence of off-target cleavage.

4) Expand the modelling with additional scenarios to include low rate release.

5) Clarify the interpretation of "bottoming out" effect in Figure 2 and if needed include statistical analysis to support it.

*Reviewer #1 (Recommendations for the authors):*

I would like to see more information about creation of the transgenic lines. Maybe each replicate of the cage trials was based on a separate insertion line? Also, maybe the fly lines are entirely homozygous.

Presumably, you still have archived samples of the experimental flies. It would really help to show that there were off target mutations with the regular Cas9 but not the high specificity Cas9.

Finally, I'd think it would really help to add some modeling of low frequency releases.

*Reviewer #2 (Recommendations for the authors):*

Overall, the manuscript is presented in a very concise way and while brevity is always appreciated, perhaps this is at the expense of clarity. In my view too much important detail is relegated to supplementary methods where inclusion in the main body of the text would almost certainly help the non-expert better understand the study.

To clarify, each of the construct lines was independently generated via Cas9-gRNA targetting EGFP. Were these sequence verified (probably not critical but good to know the insertions were identical). Why not use RMCE?

In particular, I am somewhat clear about the differences in founding populations in the cage experiments. And the details of the phenotypic assays later in the paper are a little vague in the text.

Figure 2. I am not entirely clear what the evidence for "bottoming out" is: i.e. comparing trajectories of 4 and 5, it looks to me like steady constant decline. In addition, these data suggests to me that at the highest Cas9-gRNA starting frequencies (6 and 7) there is no decline in construct levels whatsoever.

Figure 2. More clarity or better explanation of results with high frequency construct levels, particularly the difference between replicates 5 and 7.

Pg 13 Figure S4 I am not sure why one would expect heterozygotes to suffer lower of target affects, to my mind the dosage of Cas9/gRNAs is unlikely to be limiting?

*Reviewer #3 (Recommendations for the authors):*

It seems to me that, were this experiment to be designed again from the start, a useful control might be constructs that are in all respects identical but with gRNAs that target *different* intergenic, putatively non-functional genomic regions. Such constructs would help provide some insight into how much variation there is in the fitness consequences of off-target mutations, since presumably the off-target sites hit by the constructs would be different. As I mentioned in the public review, I also think any and all validation the authors could do to support their conclusion that the CasHF1 transgene has similar levels of on-target cutting as the standard Cas9 transgene would greatly support their interpretation that the loss of fitness is due primarily to off-target effects.

A little more background regarding the authors' prediction that there should be no cuts expected from the "Cas9_no-gRNAs" construct could be helpful for the naive reader. Without gRNAs, does the Cas9 protein have no endonuclease activity?

line 534: "Transformants were identified by the presence of DsRed fluorescent protein in the eyes, which usually indicated successful construct insertion". The use of "usually" here is surprising – does this indicate there were some number of flies which did not have a transgene insertion but nonetheless expressed DsRed? Is there an explanation for this finding? Similarly, "The progeny of these crosses was phenotyped for DsRed, indicating presence of a drive allele, and EGFP, indicating the presence of an intact target allele (or more rarely, a resistance allele that preserved the function of EGFP)" – how were resistance alleles that preserved the function EGFP identified? How common were such alleles?

line 606: "We compared the image-based screening pipeline to the screening method using a stereo dissecting microscope and found that the estimated genotype frequencies deviate not more than 1% from each other (n = 646 flies, 4 picture sets)." Were the deviations between the two methods biased, specifically, did the image-based method miss some fluorescent individuals that were found when screening by hand?

The methods reports a PCR-based genotyping assay, but it's not clear when this approach vs. the eye-color phenotype was used during the course of the experiments.

[Editors’ note: further revisions were suggested prior to acceptance, as described below.]

Thank you for resubmitting your work entitled "Fitness effects of CRISPR endonucleases in *Drosophila melanogaster* populations" for further consideration by *eLife*. Your revised article has been evaluated by Detlef Weigel (Senior Editor), a Reviewing Editor, and the original reviewers.

Thank you for providing a revised version of the manuscript "Fitness effects of CRISPR endonucleases in *Drosophila melanogaster* populations". We appreciated that the manuscript has been largely revised to respond to the reviewer's suggestions. We have received the report from two reviewers, and on the basis of their comments, we accept the publication of the revised version of the manuscript in *eLife*. However, both reviewers and I share the view that the authors should use more cautious language about the role of off-target to determine the fitness costs observed. The authors discuss the limitations of the study. I would propose the authors address Reviewer 1's comments about the need to discuss further (a) the lack of detection of off-target in their sequencing methods (could it be due to the approach used? Sanger seq was done on 20-30 flies. Bias in the PCR reaction for the most common allele or the limited sample size could be the reason for not detecting off-targets. Targeted amplicon seq on a larger sample size would have been a better option). (b) other factors (and likelihood) other than off-targets that could explain the observed fitness costs.

*Reviewer #1 (Recommendations for the authors):*

In my original review of the manuscript, my two main concerns were about the potential lack of use of independent fly lines and the lack of evidence regarding off-target cutting.

I am pleased to read that there were indeed many independent fly lines. It was also good to see that the authors examined the flies to determine if there were impacts on the likely genomic sites where off-target effects would be seen. Unfortunately, none of those sites showed impacts of off-target cutting. It doesn't make sense to me that given this evidence, the authors still maintain that the most likely reason for the fitness costs was off-target effects.

In the text, the authors explain that "flies that had been homozygous for the Cas9_gRNAs construct for approximately 60 generations" were used in order to detect off-target cutting at sites predicted to be the most likely to be vulnerable to such cutting. This section is very short and while the rest of the paper uses sophisticated statistical approaches to make likelihood statements, the evaluation of these negative results is just qualitative. I wonder if there is a way for the authors to use a statistical approach to ask how likely it would be that after 60 generations there would be no detected off-target cuts by their method and still be significant fitness costs.

The authors argue that there could be unpredicted off-target cutting and give one reference supporting that possibility, but in my quick reading of that paper, it still seems like there is cutting at the predicted sites. Given that the term "off-target" is used over 100 times in the text, I think these negative findings need more discussion.

The authors state that "while our maximum model comparison points to off-target fitness costs as the most plausible explanation for the observed construct frequency dynamics, the presence of off-target fitness costs does not rule out other (probably latent) factors contributing to fitness costs of CRISPR endonucleases in *Drosophila*." I would like to see a bit more discussion of these other factors and their likelihood.

*Reviewer #2 (Recommendations for the authors):*

I was happy to read the revised manuscript from the authors that, in my view, substantially improved the original submission. I found the narrative more substantive and the presentation of the ML analysis more transparent. Importantly, the limitations of the study are more clearly recognised in the text.

The recommendations from the reviewers appear to have been addressed as far as I can see, including the more technical statistical analysis.

Important data has moved from the supplement to the main text.

Taken together, I remain of the view this is an interesting and useful study.

---

## [Author Response]

Essential revisions:The results seem still premature and the manuscript will require substantial revisions. Please consider all the suggestions form the reviewers. Below is a list of essential revision requirements:

We thank the editor and all reviewers for their constructive feedback. A brief description of how we addressed these five essential revision requirements can be found below. Please see our point-by-point response for further details.

1) More details on how the transgenic lines were generated, and how they were tested, including sequencing verification of the integrity of the constructs.

We have expanded the description of how the transgenic lines were generated and tested. Most cages used independent lines that were confirmed by sequencing.

2) More detailed explanation of the experimental designs, and a clearer presentation of the data.

We have substantially reworked several parts of the manuscript to increase its clarity, focusing on a better description of the experimental design and a clearer presentation of our data and statistical analysis.

3) Provide some direct evidence of off-target cleavage.

In an attempt to directly detect some off-target cuts, we sequenced approximately 30 flies that had been homozygous for the Cas9_gRNA construct for ~60 generations at six computationally predicted off-target sites, yet found no direct evidence of cleavage at these sites. However, as we argue below and explain in detail in the revised manuscript, this does not invalidate the interpretation that the fitness costs of the Cas9_gRNAs construct were due to off-target effects at other genomic sites than those predicted. Please see our answers to point 1.d in the Public Review for further details.

4) Expand the modelling with additional scenarios to include low rate release.

We have added a new simulation analysis to study the effect of fitness reductions from off-target site cleavage on actual gene drive performance.

5) Clarify the interpretation of "bottoming out" effect in Figure 2 and if needed include statistical analysis to support it.

We have completely restructured the first part of the Results section where we had originally introduced the (admittedly vaguely defined) “bottoming out” effect. This was based only on a first visual interpretation of the cage frequency trajectories and has been removed. Instead, we now limit our descriptive text and focus more on our maximum likelihood framework with its much more rigorous statistical analysis of the cage frequency data. Please see our replies to points 2.(b) and 3.(a) for further details.

Reviewer #1 (Recommendations for the authors):I would like to see more information about creation of the transgenic lines. Maybe each replicate of the cage trials was based on a separate insertion line? Also, maybe the fly lines are entirely homozygous.

Please see our answer to point 1.a in the Public Review.

Presumably, you still have archived samples of the experimental flies. It would really help to show that there were off target mutations with the regular Cas9 but not the high specificity Cas9.

Please see our answer to point 1.d in the Public Review.

Finally, I'd think it would really help to add some modeling of low frequency releases.

Please see our answer to point 1.e in the Public Review.

Reviewer #2 (Recommendations for the authors):Overall, the manuscript is presented in a very concise way and while brevity is always appreciated, perhaps this is at the expense of clarity. In my view too much important detail is relegated to supplementary methods where inclusion in the main body of the text would almost certainly help the non-expert better understand the study.

Thanks for pointing this out. We have included details regarding both — the Cas9HF1 homing drive as well as the phenotypic assays — into the main part of the manuscript and revised the section about the maximum likelihood framework to hopefully improve clarity for the non-expert reader.

To clarify, each of the construct lines was independently generated via Cas9-gRNA targetting EGFP. Were these sequence verified (probably not critical but good to know the insertions were identical). Why not use RMCE?

Yes, each line was independently generated with CRISPR injections, allowing them to be manufactured in one step rather than two steps for RMCE. We used multiple lines to avoid any issues with defective injection methods (please also see our answer to comment 2.a in the Public Review). Though presumably less of an issue, this would still be needed with RMCE in any case to ensure the absence of line effects. As noted in the Methods and Results, the lines were screened for ability to cut the target sites. Additionally, we screened the lines for correct insertion location, the details of which are now added to the “Genotyping” section in the Methods.

In particular, I am somewhat clear about the differences in founding populations in the cage experiments. And the details of the phenotypic assays later in the paper are a little vague in the text.

Please see our answer to point 2.a, 2.d and 2.h in the Public Review.

Figure 2. I am not entirely clear what the evidence for "bottoming out" is: i.e. comparing trajectories of 4 and 5, it looks to me like steady constant decline. In addition, these data suggests to me that at the highest Cas9-gRNA starting frequencies (6 and 7) there is no decline in construct levels whatsoever.

Please see our answer to point 2.b in the Public Review.

Figure 2. More clarity or better explanation of results with high frequency construct levels, particularly the difference between replicates 5 and 7.

Please see our answer to point 2.b in the Public Review.

Pg 13 Figure S4 I am not sure why one would expect heterozygotes to suffer lower of target affects, to my mind the dosage of Cas9/gRNAs is unlikely to be limiting?

While the differences in fecundity between heterozygotes and the two homozygous genotypes were not statistically significant, there seemed to be a clear trend towards an intermediate fecundity phenotype for heterozygous individuals. We agree with the reviewer that given the reasonable assumption that Cas9/gRNAs dosage is not limiting, this observation needs further investigation in follow-up studies with larger sample sizes.

Reviewer #3 (Recommendations for the authors):It seems to me that, were this experiment to be designed again from the start, a useful control might be constructs that are in all respects identical but with gRNAs that target different intergenic, putatively non-functional genomic regions. Such constructs would help provide some insight into how much variation there is in the fitness consequences of off-target mutations, since presumably the off-target sites hit by the constructs would be different. As I mentioned in the public review, I also think any and all validation the authors could do to support their conclusion that the CasHF1 transgene has similar levels of on-target cutting as the standard Cas9 transgene would greatly support their interpretation that the loss of fitness is due primarily to off-target effects.

We agree with the reviewer that it would have been interesting to test multiple (even more than just 1 or 2) forms of the construct targeting many different intergenic regions, thus getting an idea of how much variation in fitness costs could be expected from different targets. However, going into this study, we could not predict which fitness costs we would end up finding, hence our focus on other matters. A follow-up study could certainly focus on this, but due to the time-intensive nature of cage studies, we elected to keep only the most essential controls for fitness types in this initial study.

Comparing Cas9HF1 and Cas9, we found high rates of on-target cutting in all flies sequenced when we were confirming the activities of these constructs (all flies had at least one cut target sites, and most from both lines had multiple cuts). This is now clarified in the text. A more precise comparison (albeit with a different gRNA) can be seen in our homing drive performance comparison. Cut rates were very high in both, and statistically indistinguishable from each other. Of course, we know from other studies that Cas9HF1 has intrinsically lower activity than standard Cas9, so even these findings cannot completely rule out a difference based on cut rates (though most of these differences are attributed to just a few of the gRNAs tested).

A little more background regarding the authors' prediction that there should be no cuts expected from the "Cas9_no-gRNAs" construct could be helpful for the naive reader. Without gRNAs, does the Cas9 protein have no endonuclease activity?

Thanks for pointing this out. We briefly mention now that we do not expect Cas9 to cleave DNA in the absence of gRNAs and provide two additional references in the “Construct design” section in the Results.

line 534: "Transformants were identified by the presence of DsRed fluorescent protein in the eyes, which usually indicated successful construct insertion". The use of "usually" here is surprising – does this indicate there were some number of flies which did not have a transgene insertion but nonetheless expressed DsRed? Is there an explanation for this finding? Similarly, "The progeny of these crosses was phenotyped for DsRed, indicating presence of a drive allele, and EGFP, indicating the presence of an intact target allele (or more rarely, a resistance allele that preserved the function of EGFP)" – how were resistance alleles that preserved the function EGFP identified? How common were such alleles?

In this study, all flies with DsRed did indeed have the correct construct. In our previous experiments, we had observed integration of the plasmid at other genomic sites, which would yield a DsRed phenotype, but would not be a “correct” insertion, hence our wording to note this possibility before we later exclude it. We additionally confirmed our constructs by sequencing, which is now stated in the “Genotyping” section of the Methods.

Resistance alleles that preserved target function were not distinguished from wild-type alleles in this study (the relative proportions here would be the same due to identical gRNA target sites and would not affect any comparison between Cas9 and Cas9HF1, which was the purpose of this experiment). We have adjusted the wording to avoid confusion here. It now reads:

“The progeny of these crosses was phenotyped for DsRed, indicating presence of a drive allele, and EGFP, which would usually indicate the presence of an intact target allele (or more rarely, a resistance allele that preserved the function of EGFP, though we did not attempt to distinguish these in this study).”

line 606: "We compared the image-based screening pipeline to the screening method using a stereo dissecting microscope and found that the estimated genotype frequencies deviate not more than 1% from each other (n = 646 flies, 4 picture sets)." Were the deviations between the two methods biased, specifically, did the image-based method miss some fluorescent individuals that were found when screening by hand?

For our four constructs in the cage populations (Cas9_gRNAs, Cas9_no-gRNAs, no-Cas9_no-gRNAs, Cas9HF1-gRNAs), we did not detect any non-fluorescent flies, regardless of the screening method (image-based vs. stereo dissecting microscope). Because the gRNAs do not target EGFP, but a gene-free non-heterochromatic region on a different chromosome, repair via end joining after a Cas9 or Cas9HF1 induced double strand break does not affect the eye fluorescence phenotype. Furthermore, we did not observe any systematic biases in genotype frequencies for the two screening methods.

Non-fluorescent flies did however occur in our gene drive assessment experiment in which we compared the performance of two homing drives with Cas9 and Cas9HF1 respectively (section “CasHF1 homing drive” in the Results). In this experiment, the absence of eye fluorescence (i.e., disrupted EGFP on the homologous chromosome arm) is a phenotypic indicator for non-functional r2 resistance allele presence. All flies in experiments with gene drives that were actually capable of homing were phenotyped under a stereo dissecting microscope. We have added the used phenotyping method to the section “Phenotype data analysis, Cas9HF1 homing gene drive” in our revised manuscript.

The methods reports a PCR-based genotyping assay, but it's not clear when this approach vs. the eye-color phenotype was used during the course of the experiments.

Thanks for pointing this out. The PCR-based genotyping assay was used to confirm mutant gRNA target sites (i.e, active gRNA) in progeny of individuals carrying either the Cas9_gRNAs or the Cas9HF1_gRNAs construct. The construct frequencies in the large cage populations, genotypes of individual flies in the phenotypic assays, as well as drive efficiency estimates of the Cas9HF1 homing drive are solely based on the eye-color phenotype. We now highlight this in the Results sections on “Construct design” and “Population cage experiment”, and in the Methods section on “Genotyping” of our revised manuscript.

[Editors’ note: further revisions were suggested prior to acceptance, as described below.]

Reviewer #1 (Recommendations for the authors):In my original review of the manuscript, my two main concerns were about the potential lack of use of independent fly lines and the lack of evidence regarding off-target cutting.I am pleased to read that there were indeed many independent fly lines. It was also good to see that the authors examined the flies to determine if there were impacts on the likely genomic sites where off-target effects would be seen. Unfortunately, none of those sites showed impacts of off-target cutting. It doesn't make sense to me that given this evidence, the authors still maintain that the most likely reason for the fitness costs was off-target effects.

We agree with the reviewer that at first glance, the lack of mutated alleles at the top six computationally predicted off-target sites seems to contradict our interpretation that the observed fitness costs are primarily due to off-target cleavage. However, we still believe that it is possible that off-target cleavage occurred, yet at other genomic sites than those we tested. We have extended the relevant paragraphs in the Discussion section to better guide the reader through our reasoning. It now reads:

These complexities are further underlined by the fact that we did not actually detect mutated sequences at any of the top six computationally predicted off-target sites for our Cas9_gRNAs construct. One potential explanation is that all these predicted sites in fact contain several nucleotide mismatches (often in the critical PAM-proximal 12 nucleotides), which should make cleavage less likely. These predictions by CRISPR Optimal Target Finder are based on *in vivo* cleavage specificity in cell lines where off-target cutting is known to occur more frequently than in animals (47). However, our system may be substantially different because it involves strong *in vivo* germline and early embryo expression across several generations, and it is unclear whether that would lead to higher or lower expected cleavage rates than in cell line studies.

At first glance, the lack of mutated alleles at this small set of computationally predicted sites may seem to contradict the interpretation that off-target cleavage is the primary driver of fitness costs for the Cas9_gRNAs construct. However, we believe it is likely that off-target cleavage did still occur, yet at other genomic sites than those we tested. Indeed, off-target cleavage can frequently occur at sites that are difficult to predict computationally (48, 57), and it has been observed at sites with 5 or more mismatches (49, 58, 59), meaning there are possibly thousands of potential off-target sites in the fly genome. While the individual cleavage rates at most of these sites might be vanishingly low, the combined genome-wide rate could still be substantial. Off-target sites could also simply fail to be predicted if they occur in regions of the genome that are poorly assembled. Importantly, even rare off-target cleavage events can still have a strong impact on fitness. For example, if deleterious cleavage events occur in only 10% of drive carriers but cause them to have a 50% reduction in viability or fertility, this would still amount to a net fitness cost of 5% for the construct. Such events do not need to occur consistently at the same sites to be able to substantially alter construct frequency trajectories in aggregate. For all these reasons, we believe that our inability to detect mutated alleles at a small set of computationally predicted off-target sites may in fact highlight a key benefit of our approach: It allows us to estimate fitness costs without having to rely on accurate computational predictions of putative off-target sites or requiring the very costly approach of whole-genome, population-scale sequencing.

47. S. J. Gratz, et al., Highly specific and efficient CRISPR/Cas9-catalyzed homology-directed repair in *Drosophila*. Genetics 196, 961–971 (2014).

48. J. Li, et al., Whole genome sequencing reveals rare off-target mutations and considerable inherent genetic or/and somaclonal variations in CRISPR/Cas9-edited cotton plants. Plant Biotechnol. J. 17, 858–868 (2019).

49. X. Pan, et al., Massively targeted evaluation of therapeutic CRISPR off-targets in cells. Nat. Commun. 13, 4049 (2022).

57. I. Höijer, et al., Amplification-free long-read sequencing reveals unforeseen CRISPR-Cas9 off-target activity. Genome Biol. 21, 290 (2020).

58. S. Q. Tsai, et al., GUIDE-seq enables genome-wide profiling of off-target cleavage by CRISPR-Cas nucleases. Nat. Biotechnol. 33, 187–197 (2015).

59. Y. Fu, et al., High-frequency off-target mutagenesis induced by CRISPR-Cas nucleases in human cells. Nat. Biotechnol. 31, 822–826 (2013).

In the text, the authors explain that "flies that had been homozygous for the Cas9_gRNAs construct for approximately 60 generations" were used in order to detect off-target cutting at sites predicted to be the most likely to be vulnerable to such cutting. This section is very short and while the rest of the paper uses sophisticated statistical approaches to make likelihood statements, the evaluation of these negative results is just qualitative. I wonder if there is a way for the authors to use a statistical approach to ask how likely it would be that after 60 generations there would be no detected off-target cuts by their method and still be significant fitness costs.

We thank the reviewer for pointing this out. In contrast to the analysis of the construct frequency trajectories in our cage populations, we believe that this rather straightforward analysis of presence/absence of cleaved alleles at the six computationally predicted off-target sites does not require sophisticated statistical testing. If the off-target cleavage rate at those sites would have been high enough to noticeably affect construct frequency trajectories in our cage populations in less than 20 generations, we should clearly see at least some mutated alleles in inbred transgenic fly lines after 60+ generations with our Sanger sequencing approach. The fact that we did not see any such mutated alleles (including mosaic sequences) indicates that cleavage at those six sites was unlikely to be the primary cause of the observed fitness costs.

To make this clearer, we have revised the “Evaluation of computationally predicted off-target sites” section in the Results accordingly – it now reads:

Using maximum stringency criteria to increase sensitivity, CRISPR Optimal Target Finder predicted two putative off-target sites for each of the first three gRNAs of this construct and no off-target site for its last gRNA (Supplementary File 2). These represent perhaps the most likely off-target sites in the genome (48, 49). To screen for the presence of mutations indicative of off-target cutting, we extracted genomic DNA from flies that had been homozygous for the Cas9_gRNAs construct for approximately 60 generations and performed Sanger sequencing at these six predicted off-target sites. We reasoned that if off-target cleavage at any of these six sites is responsible for the observed fitness cost (and thus occurs frequently enough to affect Cas9_gRNAs construct dynamics over the less than 20 generations of our cage experiments), then we would expect to observe at least some mutations in Cas9_gRNAs homozygotes after 60 generations.

The authors argue that there could be unpredicted off-target cutting and give one reference supporting that possibility, but in my quick reading of that paper, it still seems like there is cutting at the predicted sites. Given that the term "off-target" is used over 100 times in the text, I think these negative findings need more discussion.

The reviewer is correct – Höijer et al., did observe Cas9 cleavage at predicted off-target sites, but most critically, they found cleavage at additional sites that were not predicted. We reason in the Discussion section, that it could be possible that our flies experienced similar cleavage at such non-predicted sites as well. We have added additional references to support this notion and expanded our discussion in this area. The revised Discussion section can be found in our answer to 1.a in the Public Review.

The authors state that "while our maximum model comparison points to off-target fitness costs as the most plausible explanation for the observed construct frequency dynamics, the presence of off-target fitness costs does not rule out other (probably latent) factors contributing to fitness costs of CRISPR endonucleases in *Drosophila*." I would like to see a bit more discussion of these other factors and their likelihood.

We thank the reviewer for pointing this out. Our intention behind this section was to highlight that although the “off-target” model provides the best fit among the five selection scenarios we tested, the observed fitness costs might of course be more complex. However, as we outline in the next two paragraphs, model constrains and the limited amount of data points makes the assessment of more complex scenarios challenging. We have revised this section of the manuscript to improve clarity. Additionally, we have reduced the certainty level of our language when discussing off-target cutting throughout the manuscript.

Reviewer #2 (Recommendations for the authors):I was happy to read the revised manuscript from the authors that, in my view, substantially improved the original submission. I found the narrative more substantive and the presentation of the ML analysis more transparent. Importantly, the limitations of the study are more clearly recognised in the text.The recommendations from the reviewers appear to have been addressed as far as I can see, including the more technical statistical analysis.Important data has moved from the supplement to the main text.Taken together, I remain of the view this is an interesting and useful study.

We thank the reviewer for their thoughtful contributions, which have helped us to strengthen our manuscript.